# On the Retrieval of Sea Ice Thickness and Snow Depth using Concurrent Laser Altimetry and L-Band Remote Sensing Data

Lu Zhou[1], Shiming Xu[1], Jiping Liu[2], and Bin Wang[1,3]

[1]Ministry of Education Key Laboratory for Earth System Modeling, Department of Earth System Science, Tsinghua University, Beijing, China

[2]Department of Atmospheric and Environmental Sciences, University at Albany, State University of New York, Albany, NY, USA

[3]State Key Laboratory of Numerical Modeling for Atmospheric Sciences and Geophysical Fluid Dynamics (LASG), Institute of Atmospheric Physics, Chinese Academy of Sciences, Beijing, China

*Correspondence to:* Shiming Xu (xusm@tsinghua.edu.cn)

**Abstract.** The accurate knowledge of sea ice parameters, including sea ice thickness and snow depth over the sea ice cover, are key to both climate studies and data assimilation in operational forecasts. Large-scale active and passive remote sensing is the basis for the estimation of these parameters. In traditional altimetry or the retrieval of snow depth with passive microwave remote sensing, although the sea ice thickness and the snow depth are closely related, the retrieval of one parameter is usually carried out under assumptions over the other. For example, climatological snow depth data or as derived from reanalyses contain large or unconstrained uncertainty, which result in large uncertainty in the derived sea ice thickness and volume. In this study, we explore the potential of combined retrieval of both sea ice thickness and snow depth using the concurrent active altimetry and passive microwave remote sensing of the sea ice cover. Specifically, laser altimetry and L-band passive remote sensing data are combined using two forward models: the L-band radiation model and the isostatic relationship based on buoyancy model. Since the laser altimetry usually features much higher spatial resolution than L-band data from Soil Moisture Ocean Salinity (SMOS) satellite, there is potentially covariability between the observed snow freeboard by altimetry and the retrieval target of snow depth on the spatial scale of altimetry samples. Statistically significant correlation is discovered based on high-resolution observations from Operation IceBridge (OIB), and with a nonlinear fitting the covariability is incorporated in the retrieval algorithm. By using fitting parameters derived from large-scale surveys, the retrievability is greatly improved, as compared with the retrieval that assumes flat snow cover (i.e., no covariability). Verifications with OIB data show good match between the observed and the retrieved parameters, including both sea ice thickness and snow depth. With detailed analysis, we show that the error of the retrieval mainly arises from the difference between the modeled and the observed (SMOS) L-band brightness temperature (TB). The narrow swath and the limited coverage of the sea ice cover by altimetry is the potential source of error associated with the modeling of L-band TB and retrieval. The proposed retrieval methodology can be applied to the basin-scale retrieval of sea ice thickness and snow depth, using concurrent passive remote sensing and active laser altimetry based on satellites such as ICESat-2 and WCOM.

## 1   Introduction

Sea ice is an important factor in the global climate system, playing key roles in modulating atmosphere and ocean interaction in the polar regions, the radiation budget through albedo effects, the ocean circulation through salinity and freshwater distribution
(Screen and Simmonds, 2010; McPhee et al., 2009; Kurtz et al., 2011; Perovich et al., 2011). In the last decades, there has been rapid shrinkage of Arctic sea ice cover (Rothrock et al., 1999; Comiso et al., 2008; Stroeve et al., 2012; Laxon et al., 2013; Stocker et al., 2013), particularly in summer. In addition, the Arctic sea ice is also experiencing dramatic thinning in recent years (Kwok et al., 2009; Laxon et al., 2013), with the transition to overall younger sea ice age. Besides, the snow as accumulated over the sea ice cover also plays important roles due to its higher albedo as compared with sea ice, as well
as thermal insulation which further hinders atmosphere-ocean interaction. With respect to changes in the sea ice cover, there is also significant decrease of the snow depth over the sea ice cover in the Arctic (Webster et al., 2014) which bears great deviation from climatology (Warren et al., 1999), indicating changes in the hydrological cycles such as late accumulation due to late freeze onset. The accurate knowledge of the sea ice cover and the snow over the sea ice, is key to the understanding of related scientific questions in climate change, as well as operational usage such as seasonal forecast.

Basin-scale observation of the sea ice cover mainly relies on satellite based remote sensing. Among the various sea ice parameters retrieved from satellite data, the most established is the sea ice concentration (or coverage). Figure 1 shows the various parameters related to satellite based laser altimetry and (L-band) passive radiometry for the sea ice cover. Passive microwave remote sensing of both Arctic and Antarctic is the basis of the retrieval of sea ice extent, with near realtime coverage since about 1979 based on satellite campaigns such as Scanning Multichannel Microwave Radiometer (SMMR), the
Special Sensor Microwave/Imager (SSM/I) (Cavalieri et al., 1999), AMSR-E (Comiso et al., 2003), AMSR2 (Toudal Pedersen et al., 2017). However, the sea ice thickness is generally not retrievable through passive remote sensing techniques due to the saturation of radiative properties especially for high frequency ranges such as SMMR or SSM/I. In situ measurements of ice thickness through moored upward-looking sonar instruments and electromagnetic induction sounders mounted on sledges, ships, or helicopters/airplanes can provide sea ice thickness at specific locations or cross sections (Stroeve et al., 2014), so
they are limited in terms of spatial coverage. Active remote sensing of satellite altimetry measures the overall height of the sea surface, serving as the major approach for the thickness retrieval of the sea ice. For radar altimetry, it is usually assumed that the radar signals penetrate the snow cover, and the main reflectance plane is the sea ice-snow interface (Laxon et al., 2003, 2013). Therefore in radar altimetry, the sea ice freeboard is measured. The sea ice thickness can be retrieved under certain assumption of the snow loading, such as climatological snow depth data in Warren et al. (1999) for multi-year sea ice (MYI)
and halved for the first-year sea ice (FYI). For laser altimetry as in ICESat (Kwok and Cunningham, 2008; Kwok et al., 2009), the main reflectance surface is the snow-air interface, and the directly retrieved value is actually the snow (or total) freeboard. The snow loading is also required for the conversion of the snow freeboard to the sea ice thickness. As analyzed in Tilling et al.

(2015) and Zygmuntowska et al. (2014), the uncertainty in snow depth is the most important contributor to that of the sea ice thickness and volume.

The major reason for the uncertainty in snow depth and the loading on the sea ice cover is the lack of stable product for snow depth over the sea ice with good temporal and spatial coverage. The snow data as used in ICESat (Kwok and Cunningham, 2008) is derived from reanalysis data and satellite retrieved sea ice motion, while the climatological snow depth data in Warren et al. (1999) as used by CryoSat-2 (Laxon et al., 2013) contains large uncertainty due to interpolation and interannual variability, and may not be adequate for the present day under the context of climate change (Kwok et al., 2011; Webster et al., 2014). The retrieval of snow depth with passive microwave satellite remote sensing has been carried out in various studies. In Comiso et al. (2003), multi-band data from AMSR-E are utilized, but only for snow cover over FYI. Maaß et al. (2013b) explored the retrieval of snow depth over thick sea ice with L-band data from SMOS. SMOS provides full coverage of polar regions on a near real-time (daily) basis. It has great advantage over satellite altimetry which can only achieve basin coverage on the scale of about one month. However, the sea ice thickness is required for the retrieval. Besides, with the better penetration of L-band signal in the sea ice cover, it is also demonstrated that there is retrievability of thin sea ice thickness with L-band data, as in Kaleschke et al. (2010) and Tian-Kunze et al. (2014). Although airborne remote sensing methods have limited spatial and temporal coverage, campaigns such as NASA's Operation IceBridge (OIB) carry out high-resolution scanning of the sea ice cover (Kwok et al., 2011; Kurtz and Farrell, 2011; Kurtz et al., 2013; Brucker and Markus, 2013), and provide invaluable data that are organized into flight-track based segments of the sea ice cover. The data can be adopted for the analysis of the status and variability of the sea ice cover at fine scale, as well as basin-scale studies as in Webster et al. (2014).

In this article, we propose a new algorithm that achieves simultaneous retrieval of both sea ice thickness and snow depth, based on two observations: the L-band passive microwave remote sensing and the laser altimetry that measures the total freeboard of sea ice. The potential of retrieval of these parameters lies in that both observations (freeboard and L-band radiative properties) are determined by these sea ice parameters. Specifically, we use OIB data (sea ice thickness, snow depth and snow freeboard) and concurrent SMOS L-band brightness temperature (TB) to simulate the simultaneous retrieval. It is found that the covariability of snow depth and freeboard at the local scale can greatly affect the well-posedness of the retrieval problem, and it is crucially important to include such covariability in the retrieval algorithm. Based on both realistic retrieval scenarios and large-scale retrieval with OIB and SMOS data, we demonstrate that the proposed algorithm can simultaneously retrieve both sea ice thickness and snow depth, and the error in the retrieved parameters mainly arises from the discrepancy between the sea ice area that corresponds to the SMOS measurement and that scanned by OIB. In Section 2 we first introduce the data, the models and the protocol of the combined retrieval. Detailed statistics of snow depth and the effects of covariability is covered in Section 3. By integrating the covariability information, we propose the retrieval algorithm and carry out evaluation and analysis in Section 4. Section 5 summarizes the article and provides discussion of related topics and future work.

## 2  Data and Models

### 2.1  Data

In order to construct and evaluate the retrieval algorithm, we mainly utilize two datasets, SMOS and OIB. SMOS measures the microwave radiation emitted from the Earth's surface in L-band (1.4 $GHz$). In this article, we adopt the L3B *TB* product from SMOS. The daily gridded SMOS TB data field is generated from multiple snapshots within a day, with each snapshot involving multiple incident angles (ranging from 0 ° to 40 °) and spatially varying gain. The data is provided on the Equal-Area Scalable Earth (EASE) grid with a grid resolution of 12.5 $km$. However, due to the limitation of the satellite's antenna size, the effective resolution of L-band radiometer onboard SMOS is about 40 $km$.

High-resolution airborne remote sensing of sea ice parameters are available from OIB missions, starting in 2009 and covering western Arctic during winter months (mainly around March). This paper utilizes OIB measurements from 2012 to 2015, during which the measurements include surface temperature of the sea ice cover. The product is organized into tracks, and includes along-track measurements of total (or snow) freeboard, surface temperature, snow depth. Due to the nature of the airborne measurements, the observations are limited to a narrow swath on the order of 100 $m$. Snow freeboard products are produced from Airborne Topographic Mapper (ATM) laser altimeter (Krabill and B., 2009). Sea ice thickness is retrieved from snow freeboard (denoted $FB_s$) and snow depth (denoted $hs$), which is measured by the University of Kansas' snow radar (Leuschen, 2014). Surface temperature is determined from the IceBridge KT-19 infrared radiation pyrometer data set (Shetter et al., 2010). There is also accompanying sea ice type information, which is from EUMETSAT OSI-SAF system (Aaboe et al., 2016). Therein, the OIB Level-4 product IDCSI4 is adopted (Kurtz et al., 2013) for 2012-2013 and the remaining OIB data for 2014-2015 is from IDCSI2 Quicklook product, which is also available at NSIDC DAAC. Both of these two datasets are 40 $m$ in resolution in the along track direction.

### 2.2  Data usage protocols

Due to the difference between OIB and SMOS data in both temporal and spatial coverage, we outline the following protocols of using the two data sets. OIB and SMOS data from the same day are taken. Spatially, for each OIB flight track, we locate all the EASE grids that contain OIB measurements. Figure 2 shows a typical case. Since OIB measurements are of a small swath, we consider the OIB data (of 40 $m$ resolution) are samples of the underlying sea ice cover that contributes to a single SMOS TB measurement. However, due to the inherent resolution of SMOS is about 40 $km$ and the daily gridded field is used in this study, we approximate the correspondence of OIB and SMOS TB by considering OIB measurements in the adjacent 3×3 cells (the red segment in Figure 2 ) of equal contribution to the SMOS TB at the central cell (the one bounded by thick blue lines in Figure 2). In total, the 9 cells cover an area of about 37.5 $km$ × 37.5 $km$, which is coherent with the physical resolution of SMOS data.

It is worth noting that the area as covered by a single scan of the OIB track consists of less than 5 % of the total area that contributes to the SMOS TB. Therefore, we only treat the OIB data as samples of the underlying sea ice cover. The OIB sample count (denoted $M$) ranges from several hundreds to over 1000. The mean value of $M$ is about 700, but there exist certain areas

that are scanned more extensively, which correspond to large values of $M$. Figure A1 shows the distribution of $M$ for all available OIB data.

In order to exclude the potential effect of insufficient sampling or the inhomogeneity of the sea ice cover, we further exclude the following data for the analysis and evaluation. First, if an area is under-sampled by OIB ($M < 100$), it is not considered for further analysis. Second, we exclude the cases in which a single SMOS TB corresponds to OIB samples with different sea ice types (i.e., mixed MYI and FYI). Third, we also exclude the cases involving sea ice leads as detected by the sea ice lead map in Willmes and Heinemann (2015a) or sea ice concentration lower than 1 according to Cavalieri et al. (1996). The purpose of these treatments is to rule out the factors that may compromise the quality of the OIB samples and allow focus on the discussion of the retrieval algorithm.

The snow freeboard as measured by OIB and the SMOS TB are used as the input to the retrieval. The mean snow depth ($\overline{hs}$) and mean sea ice thickness ($\overline{hi}$) among OIB samples are used for verification of the retrieval. Besides, since we assume the underlying sea ice cover as homogeneous within the retrieval scale (within 9 cells) and treat OIB measurements as samples to it, we also use the $M$ measurements of snow depth to study the statistics of the snow depth and its covariability with snow freeboard.

## 2.3 L-band radiation model

The L-band (1.4 $GHz$) radiative property of the sea ice cover is characterized through numerical modeling based on Burke et al. (1979). The model was originally designed for the modeling of radiative transfer of the X- and L-band soil moisture. In Maaß et al. (2013b), this model is applied to sea ice and further used for the retrieval of snow depth over thick sea ice. In these works, a simple 1-layer formulation is used for both the sea ice and the snow cover over it. In order to better characterize the radiative properties of the sea ice, in this article we use a multi-layer formulation of the model with sea-ice type dependent vertical salinity and temperature profile (Zhou et al., 2017). The temperature profile in the vertical direction is linear in either the snow cover and the sea ice, assuming homogeneous thermal conductivity within the snow or the sea ice. Therefore the temperature in each sea ice or snow layer can be fully decided given the parameters of thermal conductivity, the ice bottom temperature (assumed to be -1.8 °C), and the snow surface temperature. The salinity profile of FYI differs from that of MYI. For FYI, the salinity of all layers of the sea ice all equals the bulk salinity, which decreases with the sea ice thickness. For MYI, a surface-drained profile is adopted to reflect the effect of summer melt and flushing. Figure 3.a shows the sea ice salinity profiles under the different sea ice types or thickness. The dielectric properties, the emissivity of the layers and the overall radiative properties of the sea ice cover is modeled, following Kaleschke et al. (2010) and Maaß et al. (2013b). The convergence of the modeled TB with respect to the layer count is witnessed, which is consistent with the study in Maaß et al. (2013b). In Zhou et al. (2017), it is demonstrated that the multi-layer treatment and the salinity profile MYI yields good fit between the simulated TB and SMOS TB. Appendix A covers details of the model and the verification with OIB and SMOS data. Figure 3.c and 3.d shows the modeled *TB* under typical sea ice parameters for FYI and MYI under typical winter Arctic conditions (surface temperature of -30 °C). The green contour lines are constant $FB_s$ lines. With the thickening of sea ice cover, the value of *TB* increases and saturates when $hi$ is large enough (larger than 2.5 $m$). The value of *TB* is not monotonic with respect to $FB_s$, and for certain

value combinations of snow freeboard and TB, two solutions are possible. This results in the potential problem of ill-posedness for the retrieval with realistic observational data, as is discussed in Section 3.2.

In order to match the protocol of the SMOS TB data product, we also simulate the mean of horizontal and vertical polarization *TB* among 0 ° to 40 °. We consider the correspondence between a single TB value from SMOS and the arithmetic mean of all the $M$ *TB* values simulated by the radiation model using the $M$ corresponding OIB samples (each with sea ice thickness, snow depth, surface temperature and sea ice type). Figure 3.b shows the comparison of modeled TB and SMOS TB, by using all available data. The least squares (LSQ) fit line (dashed line) and the LSQ fit line with the constraint that the slope be 1 (dotted line) are shown. The root mean square error (RMSE) in modeled *TB* as compared with SMOS data is about 3.1 $K$. The $R^2$ value for the second fit is 0.54 with an intercept of -1.637 $K$, which is treated as a model bias and canceled in further studies. As noted in Section 2.2, there is potentially insufficient sampling of OIB data, so we further consider areas with more extensive OIB sampling. Specifically, cells with large values of sample count $M$ (over 95 percentile) are considered to be more thoroughly scanned spatially, and the RMSE of *TB* for these cells drops to 1.41 $K$. Figure A2 shows the relationship between RMSE of *TB* to the value of $M$, which demonstrates that the lack of sufficient spatial coverage is an important source for the difference between the modeled *TB* and the SMOS observation. Based on the aforementioned RMSE of 1.41 $K$ for well-surveyed regions, we only consider the retrieval for cells with an *TB* error within 1.5 $K$ for further studies. In all 412 *TB* cells are available, containing 35 OIB tracks and 321'168 OIB measurements. They account for about 50% of all available *TB* cells. We consider this is a limitation of combined usage of OIB and SMOS data, and the retrieval with actual satellite laser altimetry and L-band *TB* can be free from this limitation through better altimetric scanning and wider swath as compared with OIB.

## 2.4 Isostatic equilibrium model

Apart from the L-band radiation model, the other model as used by the retrieval is the equilibrium model based on the buoyancy relationship. Under certain assumptions of the sea ice density (denoted $\rho_{ice}$), sea water density (denoted $\rho_{water}$) and snow density (denoted $\rho_{snow}$) and the equilibrium state, the sea ice thickness, snow depth and snow freeboard $FB_s$ are constrained according to Equation 1. And the sea ice thickness can be derived given the snow depth, according to Equation 2. This model is widely applied for both radar and laser altimetry for the retrieval of sea ice thickness.

$$\rho_{ice} \cdot hi + \rho_{snow} \cdot hs = \rho_{water} \cdot (hi + hs - FB_s) \tag{1}$$

$$hi = \frac{\rho_{water}}{\rho_{water} - \rho_{ice}} \cdot FB_s - \frac{\rho_{water} - \rho_{snow}}{\rho_{water} - \rho_{ice}} \cdot hs \tag{2}$$

In this study, $\rho_{water}$ and $\rho_{ice}$ are taken to be 1024 $kg/m^3$ and 915$kg/m^3$ which are derived from field measurements discussed by Wadhams et al. (1992), and $\rho_{snow}$ is 320 $kg/m^3$ derived from Warren et al. (1999).

## 3 Retrievability analysis

Under the observational constraints of *TB* and $FB_s$, both sea ice thickness and snow depth over sea ice can be retrieved (Xu et al., 2017). Figure 3.c and 3.d show *TB* as simulated by the radiation model (Section 2.3 and Appendix A) under a range of sea ice parameters. Specifically, the constant snow freeboard lines (with freeboard values) are shown. With the observed *TB* and the corresponding observation of $FB_s$, the values of sea ice thickness and snow depth can be attained through a solving process that involves the two aforementioned forward models. The theoretical retrieval problem (shown in Figure 3 ) is studied in Xu et al. (2017), with treatment of ill-posedness cases which involve two potential solutions.

For the retrieval with actual observational data, the resolution difference between the two types of observations should be accounted for. Previously in Section 2.2, we use high-resolution altimetry scans as samples for L-band passive radiometry which is of relatively coarser resolution. In this section we further analyze the statistical covariability between $hs$ and $FB_s$ on the scale of retrieval. Under the context of retrieval, we base the analysis with the freeboard measurements as a priori, and focus on how the snow depth changes with freeboard in a statistical sense. For each TB measurement, the multiple ($M$) OIB samples are subjected to statistical analysis, which shows that among these samples there exists statistically significant correlation between $FB_s$ and $hs$, which can be better characterized by a nonlinear fitting. Furthermore, the effect of the covariability on retrievability is analyzed in Section 3.2.

### 3.1 Covariability analysis based on OIB data

For the covariability between $FB_s$ and $hs$, we choose the native resolution of the OIB product ($40\ m$) as the spatial scale for analysis. Each TB corresponds to multiple ($M$) OIB samples, with each sample containing the measurement for both $FB_s$ and $hs$. We divide these samples into $FB_s$ bins, with each bin covering 5 $cm$. In total there are 30 bins, covering the range of 0 to 1.5 $m$. For samples in each bin, we compute the percentiles and the mean value of $hs$. Figure 4.a shows the mean $hs$ and the +/-1 standard deviation range and their relationship with $FB_s$, for 4 representative TB points. Furthermore, we carry out least squares linear fitting (weighted according to sample count in each bin) between $FB_s$ of the bins and the corresponding mean $hs$ in each bin. Among all available data, there exist statistically significant positive correlation between $hs$ and $FB_s$ for over 90% of all points. The values of $R^2$ are in the range of 0.06 and 0.89 (95% percentile), with the mean value of $R^2$ as 0.53. This indicates that there exists consistent covariability between snow depth and snow freeboard across Arctic sea ice cover.

However, for both FYI and MYI ice, there is saturation of the mean $hs$ with respect to $FB_s$. Besides, in the Arctic inundation is generally uncommon (i.e., $hs < FB_s$). In order to accommodate these characteristics, we propose a nonlinear fitting, as shown by Equation 3. The parameters $\alpha$ and $\beta$ are fitted according to observations. According to the equation, the value of $hs$ saturates to $\alpha \cdot \pi/2$ when $FB_s$ is large, and the value of $\alpha \cdot \beta$ (denoted $s$) which is the slope of the function at $FB_s = 0$ should be lower than 1 in order to avoid any inundation.

$$hs(FB_s) = \alpha \cdot \arctan(\beta \cdot FB_s) \tag{3}$$

Using Equation 3, the overall quality of the fitting for all available local OIB segments is improved, with mean value of $R^2$ rising from 0.53 to 0.67, and the 95% percentile of $R^2$ rises to 0.23 and 0.92 respectively. Detailed distribution of the fitted parameters for all OIB data are shown in Appendix B (Figure B1 for FYI and Figure B2 for MYI). Based on statistics of all the available OIB data, the value of $s$ for the local OIB segment is in the range of 0.49 and 0.96 (95% percentile) with a single mode distribution for both MYI and FYI (Figure B1.c and B2.c). For FYI, the mean value of $s$ is 0.71 and for MYI 0.95, which implies a generally thicker snow cover over MYI. Among all the local OIB segments, 80% of them witnessed a value of $s$ lower than 1.

Furthermore, we consider the value of $s$ to be stable across either FYI or MYI sea ice, and choose these values as universal parameters for the design of the retrieval algorithm. Figure 4.b shows fitting function of snow depth over snow freeboard based on these representative values of $s$ under various values for $\alpha$.

## 3.2 Effects of covariability on trievability

We evaluate the covariability and its effect on retrieval from several aspects. We choose 5 realistic retrieval scenarios among all the OIB and SMOS data, with two of them representing FYI retrieval, and three of them for MYI. As shown in Table 1, they represent typical retrieval problems for Arctic sea ice. Besides, the simulated TB values by the radiation model is close to the corresponding SMOS TB values (within 1.5 $K$). Based on these scenarios, we examine whether it is possible to retrieve the actual sea ice thickness and snow depth, with or without the covariability. Firstly we ignore the covariability, and assume a flat snow cover: for the $M$ OIB samples, we assume that the snow depth is uniform. For the retrieval problem, since the directly observed values are freeboard samples ($FB_s|_m$, where $m$ is the index of the samples, and $1 \leq m \leq M$), we carry out the scanning of the (uniform) snow depth $hs$ from 0 $m$ (snow free) to 1 $m$ (sufficiently deep). Under a certain value of $hs$, we retrieve the sea ice thickness $hi|_m$ for each $FB_s|_m$ with Equation 2, based on the current value of $hs$. Then the TB value for this sample ($TB|_m$) can be calculated according to the L-band radiation model, with $hi|_m$, $hs$ and surface temperature $T_{sfc}|_m$. The mean TB value is then computed as the arithmetic mean of all $TB|_m$'s, for the current value of $hs$. For any OIB sample, if the value of freeboard is smaller than the current value of $hs$, in order to avoid inundation, the snow depth for this sample is assumed to be the same as $FB_s$. If the number of samples that witness potential of inundation over 50 % of $M$, we stop the scanning even if $hs$ has not reached 1 $m$.

In order to incorporate the effect of covariability, we adopt either the global value of $s$ (0.71 for FYI and 0.95 for MYI) or the locally fitted value of $s$ (specific to each scenario), and carry out the retrieval. Figure 5.a shows 4 typical distribution of $FB_s$, and Figure 5.b shows a range of values for $\alpha$ (0 to 1) and the resulting mean value of $hs$ for the 4 typical distributions. For the range of 0 to 1, the resulting mean $hs$ covers a continuous range for each distribution. For each distribution, when $\alpha$ is very small, the corresponding $hs$ is very small for whole range $FB_s$, resulting in a very small value of mean $hs$. Furthermore, the value of mean $hs$ approaches 0 when $\alpha$ approaches 0, which in effect corresponds to "bare ice". With the grow of $\alpha$, there exists monotonous increase in the mean $hs$, and when $\alpha$ is large enough, the mean $hs$ saturates. For all of the 4 $FB_s$ distributions, we consider that the resulting mean $hs$ is reasonable for the range of $\alpha$. Therefore, the retrieval of snow depth is attained by locating the proper value of $\alpha$. Due to the potential of double solution in the retrieval, the solving of $\alpha$ is attained by a scanning

process that cover the reasonable range for $\alpha$. The scan starts from 0.001, steps by 0.01, and it is limited to a large value that yields saturation for mean $hs$. With each scanned value of $\alpha$, a corresponding value for $\beta$ can be computed as $s/\alpha$, and the snow depth $hs|_m$ for each sample can be computed with Equation 3. Then the $hi|_m$, the TB values for each sample can be computed, as well as the mean snow depth and mean TB.

We record the (mean) snow depth, and the corresponding mean TB across the scanning process. Figure 6 shows the results of scanning for the five scenarios in Table 1. Note that for the lines that represent scanning of $\alpha$ (i.e., involving covariability), the $x$-axis is the resulting values of mean $hs$, not $\alpha$. The observed TB and the simulated TB (with OIB data) are shown by solid and dot-dashed horizontal lines, respectively. Besides, the observed mean snow depth and the 50 % inundation with flat snow cover are shown by solid and dashed vertical lines, respectively. The simulated TB with flat snow cover (black dashed curve

in each subfigure) is always lower than that with covariability information (blue dashed curves for results with global $s$ and red ones for those with local $s$). For all the scenarios, the TB values that are attained through scanning can reach the observed TB with the incorporation of covariability, while with the flat snow cover assumption, the values of TB in two scenarios (III and IV) fail to reach the observation. This implies that with the flat snow cover assumption, there is no solution to the retrieval problem. We further examine the other 3 scenarios, the solutions of the retrieval problem reside at the crosspoint of the scanned

TB curves and the horizontal bars that represent observational TB values. The solutions of mean snow depth under the flat snow cover assumption are always larger than the observed mean snow depth by over 5 $cm$.

For the comparison between the covariability incorporated scanning with local $s$ and global $s$, we show that for scenario I, II, III and IV, the solutions of the two scannings are close to each other (within 2 $cm$). For Scenario II, III and IV, the solutions as produced by the scanning is close to the observed snow depth. The differences between the solutions produced by scanning

and the observed snow depth are 5 $cm$ or larger for scenario I and V, with the scanning with local $s$ produces smaller errors. It is worth noting that for the actual retrieval process, the local value of $s$ is not available, and only the global value of $s$ is usable. Lastly, for scenario III, two potential solutions exist (two crossing points between the TB scanning curve and the observational TB). Without extra observational data during retrieval, it is not possible to judge which solution is the true (or better) one. Therefore the retrieval algorithm should be able to locate both possible solutions.

The covariability as observed with OIB data plays an important role in the retrievability of the sea ice parameters. Also with OIB data, we extract the statistical relationship (Equation 3) that characterizes the covariability which can be incorporated in the retrieval. However, during retrieval, the parameter $s$ is generally not available for the local area, and the global values of $s$ (for FYI and MYI) as computed from high-resolution OIB data can be adopted.

## 4   Retrieval algorithm and evaluation

With the statistically significant covariability, we design the retrieval algorithm for sea ice thickness and snow depth that includes two distinctive phases. The overall structure of the algorithm is similar to the theoretical retrieval algorithm in Xu et al. (2017). The incorporation of covariability is further integrated, based on the nonlinear fitting in Equation 3 and the fixed value of $s$ for both FYI and MYI derived from OIB data. The first phase involves the scanning of possible snow depth

configurations. This phase is in effect carried out by the scanning of the value of $\alpha$ from 0.001 to 3 (or sufficiently large). A possible solution is detected between two adjacent values of $\alpha$, when the TB values as generated with these two values of $\alpha$ are on different sides of the observed TB. During the second phase, all the possible solutions are then attained with an iterative binary search of $\alpha$. All possible solutions are reported by the retrieval algorithm. The outline of the algorithm is presented in

Figure 7, with the two phases marked out by red and blue boxes, respectively. We also construct a reference retrieval algorithm based on the flat snow cover assumption, for which the scanning is over the snow depth instead of $\alpha$. The details of this reference algorithm is omitted for brevity.

For the typical scenarios in Table 1, we carry out the retrieval for the mean sea ice thickness ($\overline{hi}$) and the mean snow depth ($\overline{hs}$) using the standard algorithm with either the global or the local values of $s$, as well as the reference algorithm. Table

2 shows the comparison of the retrieval results and observations. The reference algorithm (with flat snow cover assumption) consistently performed worse than the standard algorithm. For scenario I and IV, it failed to attain any solution. For the standard algorithm, small error in both $\overline{hi}$ and $\overline{hs}$ is attained with the local values of $s$ specific to each scenario, as compared the retrieval with the global values of $s$. Besides, for scenario III for which two solutions are possible, the retrieval algorithm addresses both of them. The retrieval results are consistent with the retrievability analysis in Section 3.2.

We further carry out verification of the retrieval algorithm in two aspects. First, by using all available OIB data, we simulate the retrieval problem with laser altimetry measurements, and verify the retrieved $\overline{hi}$ and $\overline{hs}$ against OIB measurements. Section 4.1 covers the retrieval and analysis. Furthermore, we construct several representative retrieval scenarios in Section 4.2 and analyze the uncertainty in the retrieved parameters and carry out attribution of the uncertainty to input parameters of the retrieval.

## 4.1   Large-scale retrieval

For the systematic verification of the proposed algorithm, we carry out the retrieval with all the available OIB data (as mentioned in Section 2.3) which are from 35 OIB tracks and 412 SMOS TB measurements, and correspond to 412 retrieval cases. For each SMOS TB, the corresponding samples (snow freeboard, surface temperature and sea ice type) which are from OIB dataset are used as the input for the retrieval. The retrieval with the flat snow cover assumption (the reference algorithm) is only successful

for 50 cases, which accounts for about 12 % of available cases. For comparison, the (standard) algorithm achieves retrieval for 391 cases (95%) with the global $s$ values, and for all the TB values with the locally fit $s$ values. Figure 8 shows the comparison of retrieved mean sea ice thickness and snow depth with observations. Figure 8.a and b shows the results for $\overline{hi}$ and $\overline{hs}$, based on: (1) simulated TB (as computed from the radiation model), and (2) the local value of $s$. This represent the "idealized" retrieval problem in which there exists no extra uncertainty. As shown in Figure 8.a, the LSQ fit for $\overline{hi}$ (dash line) features a

$R^2$ value of 0.966, while the LSQ fit under the extra constraint on slope (dotted dash line) features a $R^2$ value of 0.964. For snow depth (Figure 8.b), the $R^2$ values for the two fittings are both 0.844. This indicates that the retrieval is in good agreement with the observations.

For the actual retrieval problem for which the local value of $s$ is unknown, and the observational TB values from SMOS is used, Figure 8.c and d shows the evaluation for $\overline{hi}$ and $\overline{hs}$ respectively. The fitting quality (in terms of $R^2$) for sea ice thickness

is as high as 0.89 and that for snow depth is 0.637. It is worth noting that these results are achieved with only statistical data derived from large-scale OIB surveys. Furthermore, if the retrieval is based on: (1) observed TB from SMOS, and (2) the locally fitted value of $s$, the $R^2$ values for the fitting are 0.91 and 0.65 for sea ice thickness and snow depth respectively, with virtually no change in the fitting lines (not shown). There is minor increase in quality (0.91 versus 0.89 and 0.65 versus 0.637)

and a relatively large gap to the "idealized" retrieval. As a comparison, we also carry out retrieval with the TB with forward model and the local values of $s$, and the $R^2$ for fittings between the retrieved and the observed parameters for sea ice thickness and snow depth are 0.96 and 0.84, respectively. This indicates that the difference (or error) of the modeled and the observed TB plays an important role in affecting the quality of the retrieval. The discrepancy between the observed TB and the modeled TB may arise from several factors: (1) the imperfect radiation model, including its formulation as well as the model parameters;

(2) the mismatch between the altimetry scans and L-band passive observations, as introduced in Section 2.2. The areas with more extensive OIB scans are shown of lower TB error (see Figure A2), indicating that the error in the retrieved parameters can be potentially reduced with better altimetry coverage.

For comparison, we also carry out the retrieval which only involves TB and the mean value of $FB_s$. This retrieval problem ignores the resolution difference between altimetry scans and L-band radiometry, and generally corresponds to the theoretical

retrieval problem analyzed in Xu et al. (2017). Specifically, for the use of OIB data, the mean value of $M$ samples of $FB_s$ is computed, and further combined with TB for the retrieval of a single value for both $hi$ and $hs$. Since only the mean $FB_s$ is involved in the retrieval, covariability does not play a role in the retrieval. By using the same SMOS and OIB data as the evaluation in Figure 8, the retrieval yields $R^2$ of 0.78 and 0.50 for $hi$ and $hs$ (fitting between the retrieved and the observed parameter). For comparison, under the realistic retrieval results (Figure 8.c and d), the quality of retrieval is much improved

for both $hi$ ($R^2$ from 0.78 to 0.89) and $hs$ ($R^2$ from 0.50 to 0.64). This demonstrates that the high-resolution altimetry samples and the accompanying covariability information play an important role in improving the quality of the retrieval.

Based on the retrieval with large-scale observational data, the proposed algorithm achieves effective retrieval of both sea ice thickness and snow depth, by using simultaneous remote sensing of the sea ice cover, i.e., laser altimetry and L-band passive microwave sensing. The statistics of snow depth and its covariability with snow freeboard on the spatial scale of retrieval play

an important role in improving the well-posedness of the retrieval problem, as well as the quality of the retrieved parameters.

### 4.2   Uncertainty analysis

In order to assess the uncertainty of the retrieved parameters, we further design four realistic retrieval scenarios from OIB and SMOS data listed in Table 3.a. Due to the nonlinear relationship between sea ice parameters and *TB*, we cannot directly compute the uncertainty in $hi$ or $hs$. Instead, Monte-Carlo (MC) simulation is adopted. For each scenario in Tabel 3.a, four

sets of MC simulations are constructed, each containing: (1) random perturbations to *TB* only, (2) random perturbations to $FB_s$ only, (3) random perturbations to $s$ only, and (4) random perturbations to *TB*, $FB_s$ and $s$ altogether. Each set contains 1000 random sampling to these parameters.

The perturbations to *TB* follow Normal Distribution and SMOS dataset (in terms of standard deviation of the uncertainty). The perturbations to the $M$ values of $FB_s$ are based on OIB data specification and follow Log-Normal Distribution. The

perturbations to $s$ are specific to sea ice type (FYI or MYI), and based on the statistics of $s$ as derived from all OIB data. As shown in Appendix B, the distribution of $s$ can be well characterized by Beta Distribution for both FYI and MYI. The fitting to Beta Distribution is then carried out for both FYI and MYI according to Equation 4, where $a$, $b$ and *const* are fitted parameters by using OIB data at 40 $m$ resolution (see Figure B3). For FYI, $a$, $b$, and *const* are 4.31, 2.00 and 1.00 respectively, and for MYI, 4.25, 2.06 and 1.2.

$$f(x|a, b, constant) = \frac{const}{B(a,b)} x^{(a-1)} (1-x)^{(b-1)} \tag{4}$$

The perturbations to $s$ follow the fitted Beta Distribution. Furthermore, the perturbations to *TB*, $FB_s$ and $s$ are treated as independent. Each MC simulation (of 1000 simulations) contains a set of perturbed input parameters, and corresponds to a retrieval problem. Based on the results from the 1000 simulations, the uncertainty of the retrieved $hi$ and $hs$ are computed by biased standard deviation estimation with respect to the original retrieval which involves no perturbation. Table 3.b shows the relative uncertainty of $hi$ and $hs$ for each experiments for all scenarios. First, the relative uncertainty for $hi$ or $hs$ is at most about 25%. Also, all scenarios show that $s$ plays a minor role in terms of uncertainty, as compared with *TB* or $FB_s$. *TB* and $FB_s$ play a comparable role in the uncertainty of the retrieved parameters. Moreover, for both FYI and MYI, the uncertainty in the retrieved $hi$ and $hs$ is relatively lower for thicker ice and deeper snow cover. The uncertainty of *TB* (or $FB_s$) is not correlated spatially, and that of $s$ is based on basin-scale statistics from OIB. Therefore, the uncertainty of the retrieved $hi$ (or $hs$) is not spatially correlated, resulting in effective reduction of the uncertainty in the sea ice volume (or snow volume).

## 5 Summary and discussion

In this study, we introduce a new algorithm for retrieving multiple Arctic sea ice parameters based on combination of L-band passive microwave remote sensing and active laser altimetry. Two physical models, the L-band radiation model and the buoyancy relationship, are adopted to constrain the sea ice thickness and snow depth. They are used as forward models during an iterative retrieval process that solves the sea ice parameters that satisfy the observed L-band TB and snow freeboard values. Specifically, according to high-resolution observations, there exists statistically significant covariability between the snow depth and the snow freeboard. This information of covariability is further incorporated in the retrieval algorithm, and it is demonstrated that the covariability plays a key role in the retrievability. Specifically, a nonlinear fitting that characterizes the covariability is derived from OIB data, and a parameter (initial slope of the fitting function) is considered invariant for FYI and MYI, and further adopted by the retrieval algorithm. Verification with available OIB data shows that both sea ice thickness and snow depth are retrieved, with the error in both parameters mainly arising from the mismatch between modeled and observed TB values. This algorithm can be applied to the large-scale retrieval of sea ice thickness and snow depth using concurrent L-band satellite remote sensing and satellite altimetry of the sea ice cover such as Abdalati et al. (2010).

**Difference with existing retrieval algorithms:**

In traditional (laser) satellite altimetry, the retrieval of sea ice thickness mainly relies on (adapted) climatological snow depth or data as derived from reanalyses, which may contain unconstrained uncertainty due to model biases as well as missing

physical processes. Besides, these snow depth data usually lack fine-scale details that match the resolution of satellite altimetry, such as the covariability characteristics. On the other hand, the retrieval of snow depth using L-band SMOS data as in Maaß et al. (2013b) relies on the a priori knowledge of the thickness of the (thick) sea ice. Contrary to these existing retrieval algorithms, the proposed algorithm carries out retrieval of both sea ice thickness and snow depth, with the concurrent active and passive remote sensing of the sea ice cover. Since no climatological snow depth or any other derived snow data is used in the algorithm, the retrieved sea ice thickness do not suffer from the potential lack of efficacy of these data.

**Covariability analysis:**

In Kwok et al. (2011), statistical analyses are carried out between snow depth and snow freeboard, which also show covariability between the two. As also noted in Kwok et al. (2011), the derivation of these two parameters are measured with independent instruments by OIB. The statistically significant relationship as represented by covariability is due to physical processes relating to the snow loading and its effects on the total freeboard. However, it is worth noting that the scale and the resolution as adopted in Kwok et al. (2011) are about 400 $km$ and 4 $km$, respectively. They are both much larger than the those as used in this study (about 40 $km$ and 40 $m$). While the analysis in Kwok et al. (2011) is carried out on coarser spatial scales, our work focuses on the spatial scale that is relevant to the retrieval of sea ice parameters. We demonstrate that on this relatively small spatial scale, there still exists covariability between snow depth and snow freeboard.

**Uncertainty estimation related to model parameters:**

Besides the input parameters to the retrieval (*TB*, *FB$_s$* and *s*), model parameters also play an important role in modulating the uncertainty for retrieval (Zygmuntowska et al., 2014; Xu et al., 2017). For this study, we adopt constant parameters for density values following protocols of OIB, mainly for the direct comparison with OIB dataset. However, their effect on the uncertainty of retrieved parameters should be accounted for in a systematic approach, similar to Xu et al. (2017). Monte-Carlo simulations can be adopted for the quantification of the uncertainty, through perturbations to both input and model parameters.

**Outlook to satellite based retrieval:**

The proposed retrieval method is the basis for the retrieval of sea ice parameters with data from concurrent satellite campaigns. Although there was no concurrent L-band satellite observation with the ICESat campaign, there are candidate satellite campaigns such as WCOM (Shi et al., 2016) that provides concurrent L-band observation with the planned ICESat-2 campaign. For the study with satellite data, there exist several practical issues. First, the snow surface temperature is provided by airborne sensors in OIB, but not generally available with laser altimetry. Several data sources serve as candidate data for the concurrent surface temperature field, such as reanalysis data (Dee et al., 2011), MODIS based product (Hall et al., 2004). Second, there exists small-scale variability of the sea ice cover such as leads, which were not considered for the analysis and verification in this study. As shown in Zhou et al. (2017), the presence of sea ice leads has profound effect in lowering the overall TB on the scale of SMOS observations. Leads can be treated as small-scale heterogeneity of the sea ice cover, and the incorporation of lead maps such as Willmes and Heinemann (2015b) effectively reduces the overestimation of TB, as studied in Zhou et al. (2017). Specifically, the lead map can be adopted by the retrieval through the integration with the forward radiation model. Other types of small scale variability such as mixture of FYI and MYI, should be also accounted for using sea ice type maps. Third, the covariability explored in this study is on the spatial scale of the original OIB data (i.e., 40 $m$). For each specific satel-

lite altimetry, we consider the freeboard measurement the mean freeboard value within a certain spatial range. For ICESat-2, each laser scan dot covers a circular region of about 70 $m$ in diameter (Abdalati et al., 2010). The scaling of the covariability should be studied for the specific resolution of the satellite altimetry. By using 70 $m$ as the typical resolution of ICESat-2, we deduce the value of $s$ at this resolution by manual coarsening OIB's data by averaging adjacent points. In effect, the value of $s$

at 80 $m$ is computed, which shows a slight decrease of $s$ for both FYI and MYI. Figure B3 shows the general scaling of $s$ for the resolution range from 40 $m$ to 240 $m$. Fourth, in order to estimate the uncertainty of the retrieved parameters, the effects of surface temperature, as well as other data sources (including TB, freeboard measurements, and the value of $s$), should be evaluated in a systematic way. Due to the nonlinear relationship between TB and the sea ice parameters, Monte-Carlo simulations can be carried out for the quantification of the uncertainty. Besides, for the historical data from ICESat (Kwok and

Cunningham, 2008) during the first decade of the 21st century, due to the lack of basin-scale L-band observation for the Arctic, other passive remote sensing data such as C-band data from AMSR-E can be exploited in a similar manner for the retrieval of these historical data.

The native spatial resolution of AMSR-E based C-band remote sensing product is over 60 $km$, which is coarser than that of SMOS L-band data, but provides similar, daily coverage for the Arctic. Therefore, the resolution difference between AMSR-E

based C-band data products and ICESat data should be accounted for in a similar approach as in Section 2.2. Besides, due to the relatively shorter wavelength of C-band as compared with L-band, the penetration depth of C-band signal in sea ice cover is potentially shallower, resulting in more premature saturation of C-band signal to sea ice thickness. Under the assumption of a uniform and dry snow cover, the relatively long wavelength of C-band and L-band ensures that the snow cover is "transparent" to the L-/C-band signal. For L-band and C-band, and there exists good potential for retrieval through the thermodynamical

modulation of the sea ice thickness by the snow cover, as indicated by Maaß et al. (2013a).

The L-band radiation model as adopted by this article can also be used for the concurrent retrieval of sea ice parameters with L-band passive radiometry and radar altimetry (RA). While laser altimetry with ICESat covered the historical era in the 2000's, CryoSat-2 based RA is an ongoing campaign which started in early 2010's and overlaps with existing L-band and C-band passive campaigns, including SMOS, SMAP, and AMSR-2. According to the theoretical study in Xu et al. (2017), the

retrieval that combines RA with L-band data is potentially free from the ambiguous solutions present in this study. Besides, there also exists resolution differences between RA (e.g., 300 $m$ for CryoSat-2) and L-band data such as SMOS. Measurements from RA can be treated as high-resolution sampling of the sea ice area that correspond to a single L-band TB. Furthermore, based on the analysis and methods proposed in this article, the covariability between snow depth and sea ice freeboard can be further incorporated in the combined retrieval with RA and L-band passive remote sensing data.

*Acknowledgements.* This work is partially supported by National Key R & D Program of China under the grant number of 2017YFA0603902 and the General Program of National Science Foundation of China under the grant number of 41575076. The authors would like to thank the editors and referees for their invaluable efforts in improving the manuscript. SMOS data is provided from Integrated Climate Data Center (ICDC), icdc.cen.uni-hamburg.de, University of Hamburg, Germany, Digital media. http://icdc.cen.uni-hamburg.de/1/daten/cryosphere/

l3b-smos-tb.html. [Data Accessed: 2017/10/25]. OIB and SSM/I sea ice concentration data are provided by NASA National Snow and Ice Data Center Distributed Active Archive Center, Boulder, Colorado USA. doi: http://dx.doi.org/10.5067/7XJ9HRV50O57. [Data Accessed: 2017/10/25]. Besides, the authors are grateful to Willmes, S. and Heinemann, G. for the provision of Arctic sea ice lead map.

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

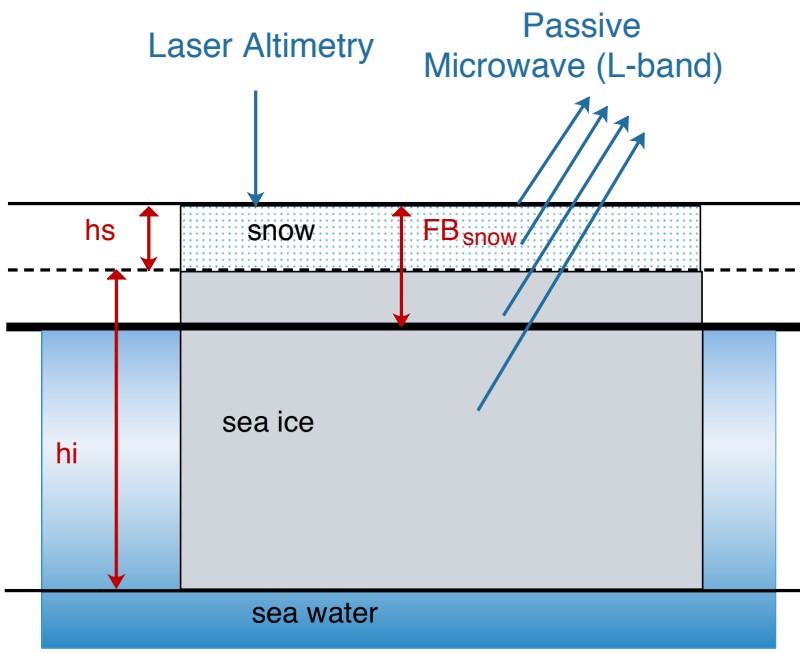

**Figure 1.** Sea ice parameters in the active and passive remote sensing of the sea ice cover, including sea ice thickness ($hi$), snow depth ($hs$) and snow freeboard ($FB_s$).

Xu, S., Zhou, L., Liu, J., Lu, H., and Wang, B.: Data Synergy between Altimetry and L-Band Passive Microwave Remote Sensing for the Retrieval of Sea Ice Parameters - A Theoretical Study of Methodology, Remote Sensing, 9, doi:10.3390/rs9101079, 2017.

Yu, Y. and Rothrock, D.: Thin ice thickness from satellite thermal imagery, Journal of Geophysical Research: Oceans, 101, 25 753–25 766, 1996.

Zhou, L., Xu, S., Liu, J., Lu, H., and Wang, B.: Improving L-band radiation model and representation of small-scale variability to simulate brightness temperature of sea ice, International Journal of Remote Sensing, 38, 7070–7084, doi:10.1080/01431161.2017.1371862, http://dx.doi.org/10.1080/01431161.2017.1371862, 2017.

Zubov, N. N.: Arctic ice, Tech. rep., NAVAL OCEANOGRAPHIC OFFICE WASHINGTON DC, 1963.

Zygmuntowska, M., Rampal, P., Ivanova, N., and Smedsrud, L. H.: Uncertainties in Arctic sea ice thickness and volume: new estimates and
implications for trends, The Cryosphere, 8, 705 – 720, doi:10.5194/tc-8-705-2014, www.the-cryosphere.net/8/705/2014/, 2014.

**Appendix A: L-band radiation model**

The L-band (1.4 *GHz*) radiation model as used for retrieval describes the radiation emitted from snow covered sea ice that floats over sea water. The model was originally developed for soil moisture applications in Burke et al. (1979), and further adopted for sea ice in Maaß et al. (2013b). As introduced in Zhou et al. (2017), improvements to the model are made to better
characterize the vertical structure of the sea ice that is specific to each sea ice type. Details are provided below.

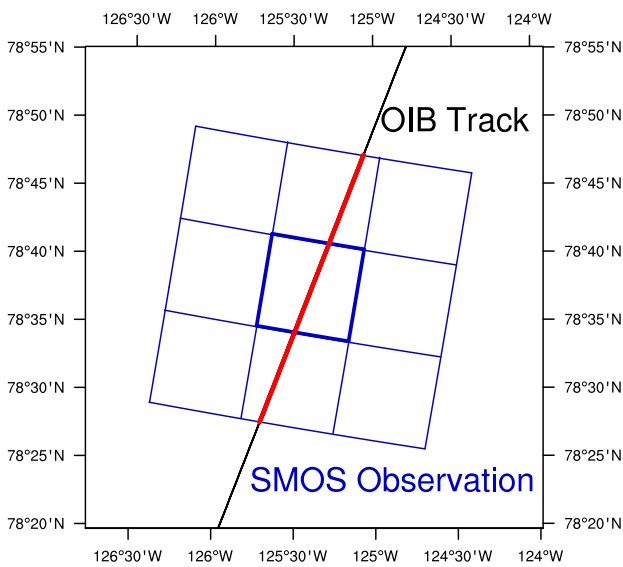

**Figure 2.** Data match between OIB and SMOS data. SMOS TB product is provided on the 12.5 $km$ EASE grid (shown by blue rectangular cells). However, the inherent resolution of SMOS TB is of about 40 $km$. The red/black line represents the OIB track. Therefore, in order to accommodate the resolution differences, OIB samples that reside within the 9 cells (red) are considered to be of equal contribution to the TB value at the central EASE grid cell (outline by the thick blue line).

## A1 General information

The modeling of the radiative properties of the sea ice cover include 4 types of media in the vertical direction: the sea water beneath the sea ice, the sea ice, the snow cover over the sea ice, and the air. The sea water (air) are considered to be semi-infinite beneath (above) the sea ice cover. The sea ice is further divided into $N$ layers in the vertical direction, with each layer of the same height. For the snow cover, a homogeneous structure is assumed, with prescribed parameters such as thermal conductivity and permittivity. Besides, a dry snow cover is assumed, and snow morphology features (such as differentiation between wind slab and depth hoar) and other vertical structures are not considered. The (SMOS) observed brightness temperature (TB) is assumed to be the multi-angle mean (0-40 degrees) TB as radiated from the aforementioned multi-layer media.

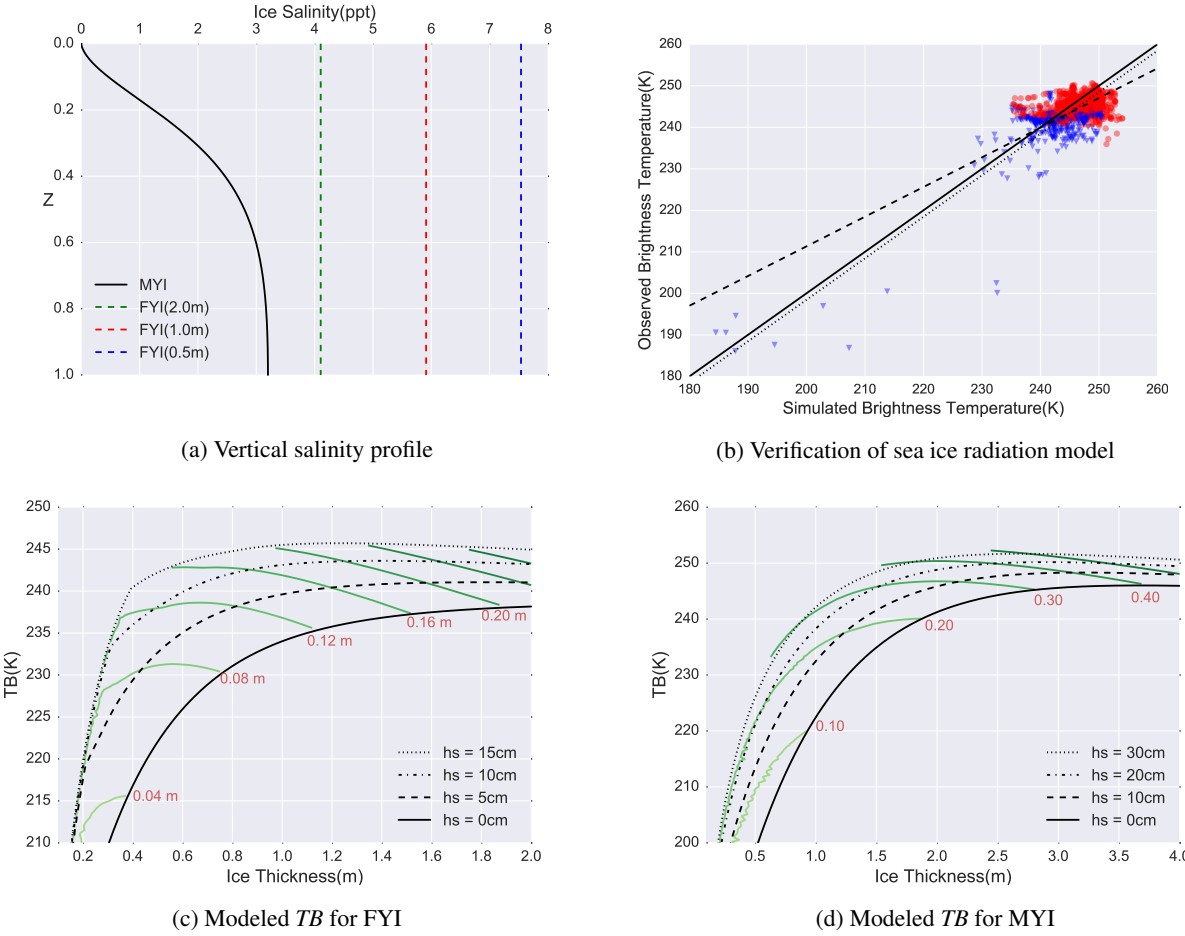

(a) Vertical salinity profile

(b) Verification of sea ice radiation model

(c) Modeled *TB* for FYI

(d) Modeled *TB* for MYI

**Figure 3.** L-band radiation model. Subfigure a shows sea ice salinity profile for FYI (dotted lines) and MYI (solid line). The vertical axis (Z) is normalized with respect to the sea ice thickness. The comparison of the simulated TB based on OIB data and the observed SMOS TB is presented in subfigure b. Blue triangles represent FYI, while red circles MYI. The dashed (dotted) line is the least squares fit (least squares fit under the constraint that slope be 1). The Root Mean Square Error of TB is 3.1 $K$. Subfigure c and d show the modeled TB under typical sea ice parameters ($hi$ and $hs$), assuming Arctic winter conditions (surface temperature of -30 $^\circ C$). The green lines represent constant snow freeboard lines.

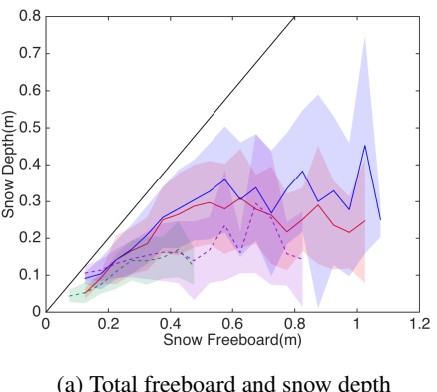

(a) Total freeboard and snow depth

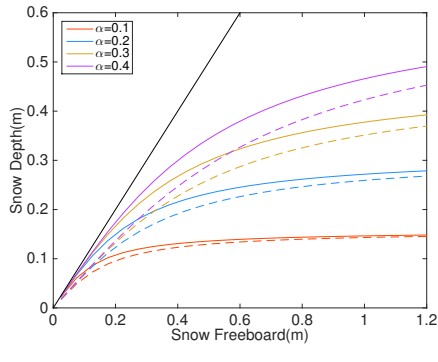

(b) Functional fitting for snow depth

**Figure 4.** Statistics of snow depth from OIB at the local scale of retrieval. Subfigure a shows the mean and the $+/-1$ standard deviation of the snow depth within each snow freeboard bin (from $0\ m$ to $1.5\ m$ by the interval of $5\ cm$), shown by lines and shaded areas for 4 realistic cases of OIB. Subfigure b shows the the nonlinear fitting of snow depth over snow freeboard (Equation 3) under representative $s$ values (0.71 for FYI and 0.95 for MYI) and various values of $\alpha$. Solid color lines are for MYI and dashed ones for FYI. The solid black line is $y = x$.

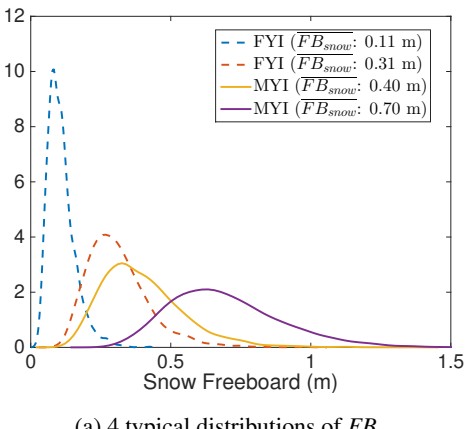

(a) 4 typical distributions of $FB_s$

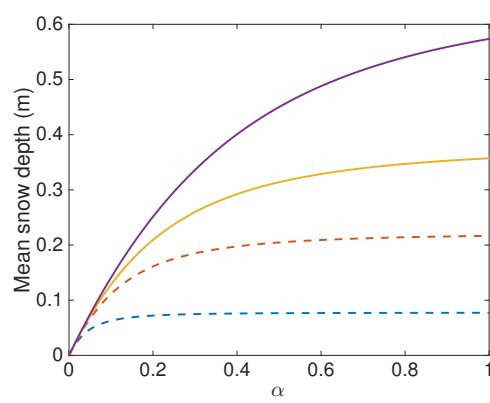

(b) Derived mean $hs$ with respect to the values for $\alpha$

**Figure 5.** Typical distributions of $FB_s$ and the range of mean $hs$ for $0 < \alpha < 1$. Subfigure a shows the 4 distributions (2 for FYI and 2 for MYI) and the corresponding mean value of $FB_s$. Global values of $s$ for FYI and MYI are adopted. For these 4 distributions, subfigure b shows the mean $hs$ for the range of $\alpha$ between 0 and 1. Mean $hs$ increases monotonically with $\alpha$, and saturates when $\alpha$ is large.

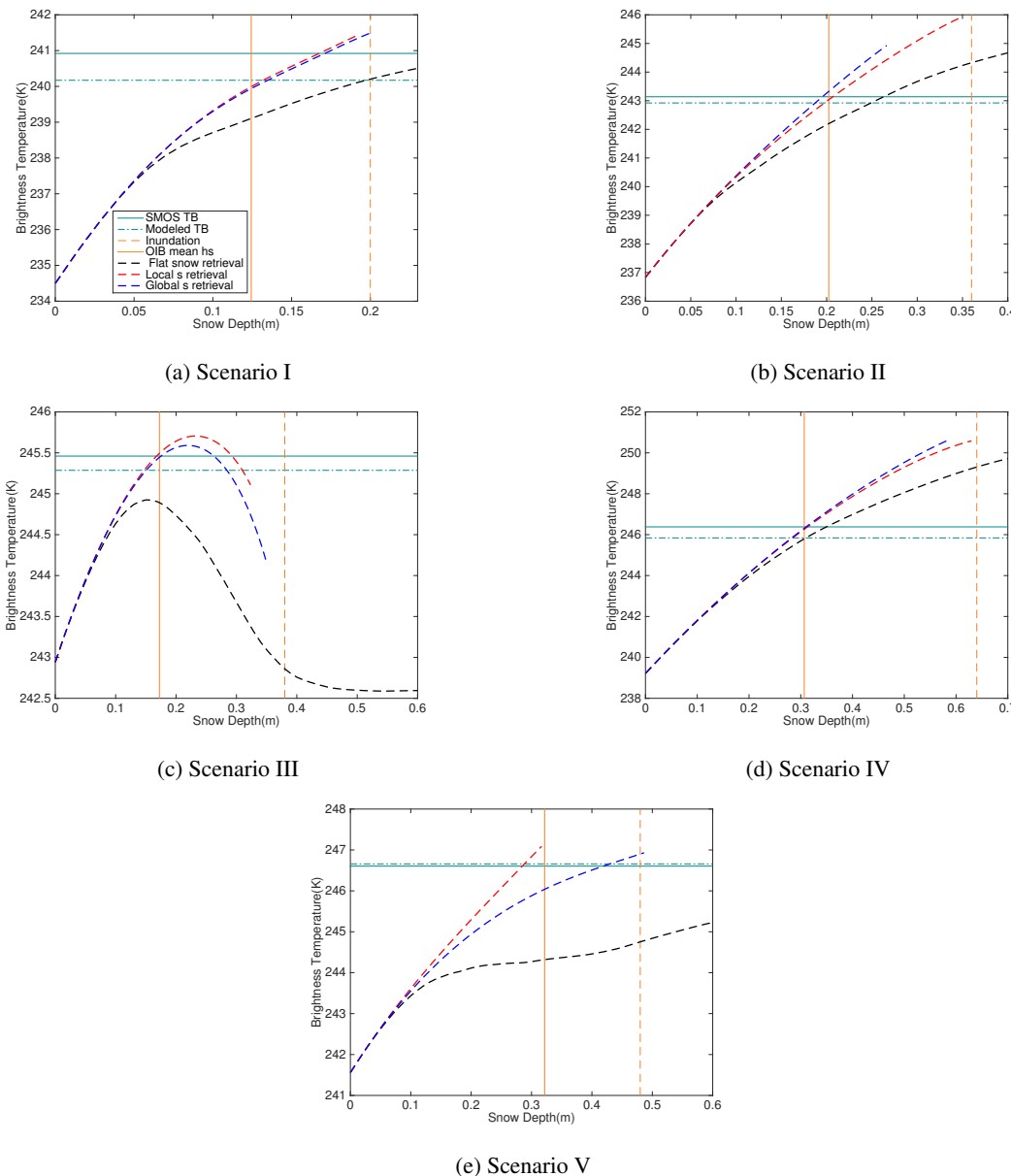

**Figure 6.** Retrievability study with different retrieval scenarios. The horizontal solid (dotted-dashed) lines are the SMOS (modeled) *TB*. The vertical solid lines represent the values of the mean snow depth from OIB observation. The black dashed curves denote the values of *TB* generated by scanning of $hs$ under the flat snow cover assumption, and the vertical dashed lines denote the values of $hs$ that result in 50% OIB samples to be inundated. The red (blue) dashed curves (with the corresponding mean snow depth) are the values of *TB* generated by scanning of $\alpha$ with the local (global) values of $s$ as in Equation 3.

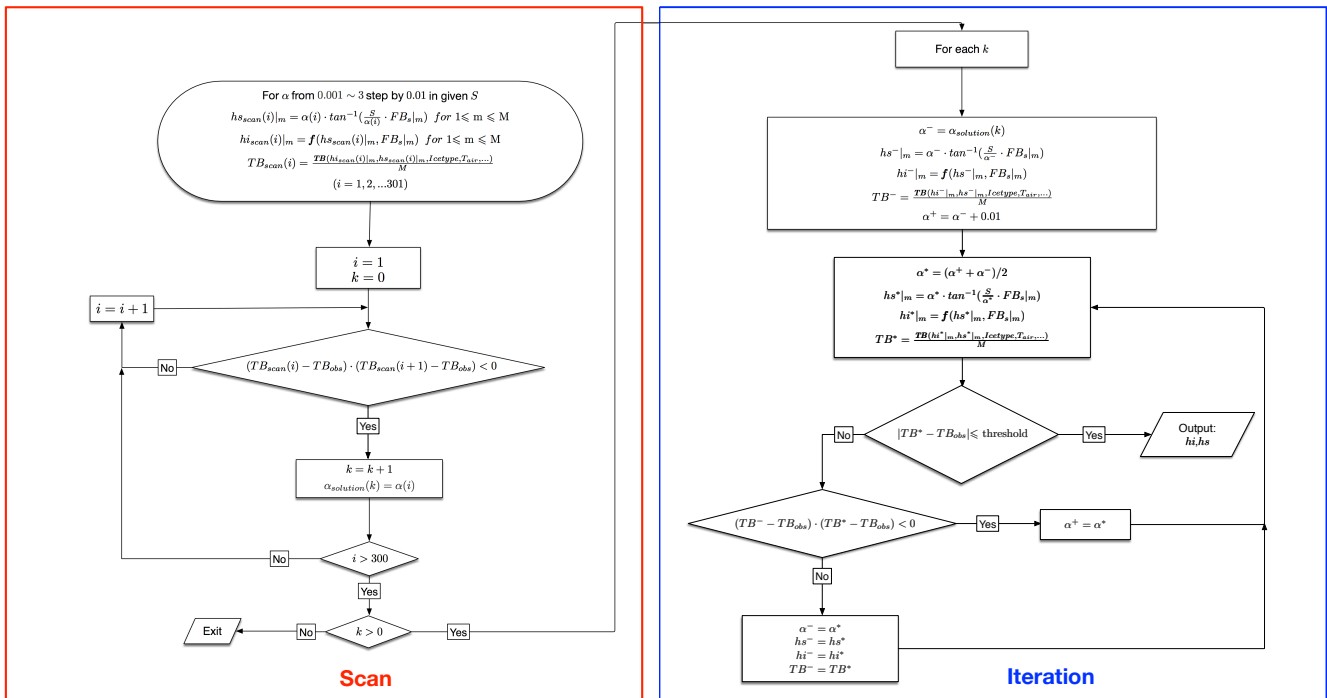

**Figure 7.** Flow chart for retrieval algorithm. Two phases are marked out. The red box includes the scanning process for the potential solutions to the retrieval problem, and the blue box the iterative binary search for the solving process.

## A2 Temperature and salinity structure

The radiation model characterizes the vertical structure of the sea ice cover, by specifying the temperature and salinity of each layer based on the sea ice type and the snow surface temperature ($T_{surf}$). The vertical temperature profile is determined by the overall thermal condition as defined by $T_{surf}$ and the thermal conductivity for sea ice ($k_{ice}$) and that of snow ($k_{snow}$). The

5   bottom of the sea ice is assumed to be at freezing temperature of -1.8 °C (denoted $T_{water}$). Based on observation-based fittings in Untersteiner (1964) and Yu and Rothrock (1996), $k_{ice}$ and $k_{snow}$ are defined as follows.

$$k_{ice} = 2.034 W m^{-1} K^{-1} + 0.13 W kg^{-1} m^{-2} \frac{S_{ice}}{T_{ice} - 273.15}$$

$$k_{snow} = 0.31 W m^{-1} K^{-1}$$

In this study, we consider the change of $k_{ice}$ within the sea ice of minor effects, and use a bulk value for $k_{ice}$, resulting in a linear temperature profile within the sea ice. This bulk value is determined by the bulk value of $S_{ice}$. The temperature profile

10   is assumed to be continuous through the media interfaces, and ice temperature is assumed to equal the snow temperature at the snow-ice interface. Given $T_{surf}$ based on other observations (such as MODIS), the bulk ice and snow temperatures $T_{ice}$ and $T_{snow}$ can be written as follows ($K = (k_{snow} hi + k_{ice} hs)^{-1}$).

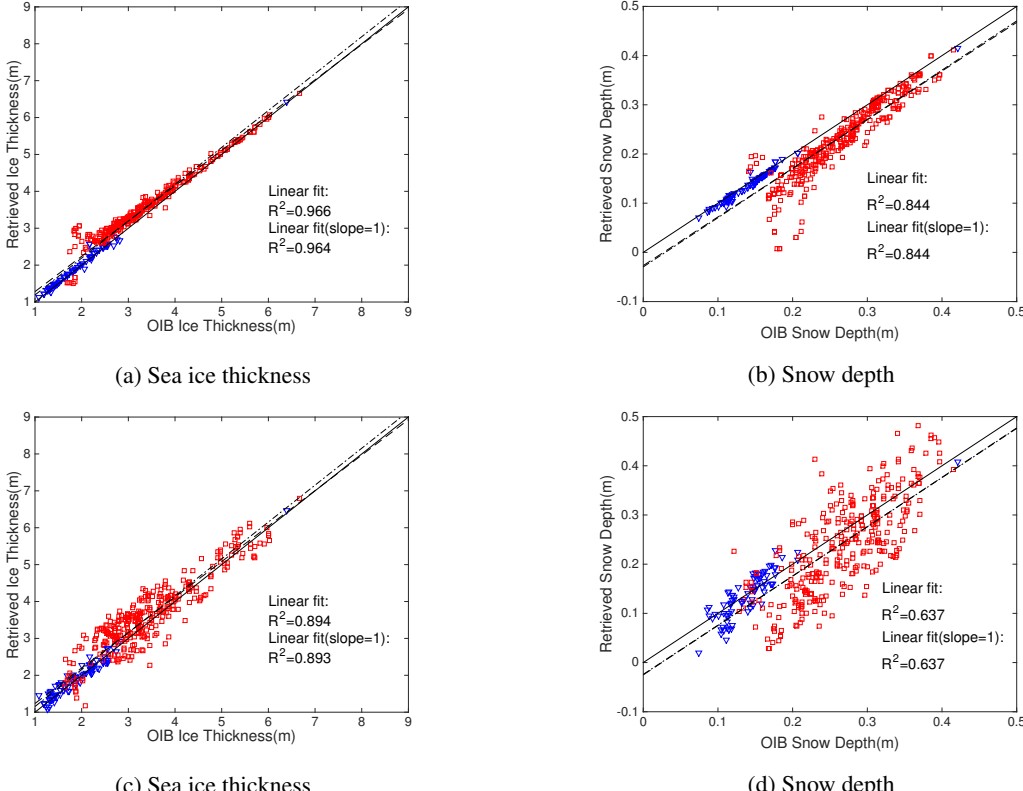

(a) Sea ice thickness

(b) Snow depth

(c) Sea ice thickness

(d) Snow depth

**Figure 8.** Large-scale retrieval of mean sea ice thickness (subfigure a and c) and mean snow depth (subfigure b and d) and verification with OIB observations. In each subfigure: blue triangles (red rectangles) denote FYI (MYI); the solid line is the 1:1 line; the dashed (dashed dotted) line represents the linear fitting (linear fitting line with the constraint that the slope be 1). The quality of fittings in terms of $R^2$ are also shown. Subfigure a and b represent the comparison results for the retrieval with modeled TB and the local values of $s$. Subfigure c and d represent the results with SMOS TB and the global values of $s$ as derived from OIB data.

$$T_{ice} = T_{water} + \frac{1}{2} K (T_{surf} - T_{water}) k_{snow} hi$$

$$T_{snow} = \frac{1}{2} (T_{water} + T_{surf} + K (T_{surf} - T_{water}) k_{ice} hs)$$

Since a bulk value is adopted for both $k_{ice}$ and $k_{snow}$, given any $T_{surf}$, the temperature profile is linear within the snow cover, as well as the sea ice. Then, the temperature of each layer of the sea ice cover can be computed.

For the salinity, sea ice type is considered with differentiation between MYI and FYI. For FYI, the salinity is assumed to be homogeneous in the vertical direction, and equals the bulk salinity as prescribed by the sea ice thickness. The bulk salinity for FYI is in turn adapted from the multi-linear structure in Cox and Weeks (1974), and defined as follows (where the ice salinity, denoted $S_{ice}$, is in ppt).

**Table 1.** Typical scenarios for retrievability studies. The mean sea ice thickness ($\overline{hi}$), mean snow depth ($\overline{hs}$), mean snow freeboard ($\overline{FBs}$), observed TB from SMOS and the simulated TB from forward radiation model are shown. Scenario I and II are FYI, and scenario III, IV and V are MYI.

| Ice type | Scenario | $\overline{hi}$ (m) | $\overline{hs}$ (m) | $\overline{FB_s}$ (m) | TB (K) | |
|---|---|---|---|---|---|---|
| | | | | | Simulated | Observed |
| FYI | I | 1.28 | 0.12 | 0.2212 | 245.84 | 246.38 |
| | II | 2.25 | 0.20 | 0.3790 | 242.92 | 243.14 |
| MYI | III | 2.46 | 0.17 | 0.3807 | 245.29 | 245.46 |
| | IV | 3.01 | 0.32 | 0.5419 | 246.66 | 246.61 |
| | V | 4.13 | 0.31 | 0.6509 | 245.84 | 246.38 |

$$S_{ice} = 6.08 * e^{(-5.81*hi)} + 7.409 * e^{(-0.5228*hi)}$$

With the deepening of the FYI sea ice cover, the bulk salinity decreases, and its minimum value is kept above 1.5 ppt. On the other hand, for MYI, in order to reflect the effect of brine drainage and flushing during the melt season, a vertical salinity profile is adopted following Schwarzacher (1959). For the $k$-th sea ice layer ($k = 0$ for the surface layer of the sea ice), the mean salinity ($S_{i,k}$) is prescribed as:

$$S_{i,k} = \frac{1}{2} S_{max}[1 - cos(\pi z^{a/(z+b)})]$$

A normalized vertical coordinate ($z$) is adopted with respect to sea ice thickness, starting from $z = 0$ for the ice surface to $z = 1$ for the bottom of the ice. For layer $k$, $z = (k - 1/2)/N$ and the corresponding salinity of the layer can be computed. $N$ is the total number of ice layers, and $S_{max}$=3.2 ppt, $a$=0.407, $b$=0.573 which are the fitted parameters from in-situ observations of MYI salinity. Therefore, for MYI, the sea ice salinity ranges from 0 at the top of the surface ($z = 0$) to $S_{max}$ at the bottom ($z = 1$). The sea water salinity is fixed at constant 33 $g/kg$.

**Table 2.** Retrieved results ($\overline{hi}$ and $\overline{hs}$, in units of meters) for five scenarios under different retrieval algorithms. In scenario II, IV and V, the retrieval with flat snow cover assumption is unsuccessful. The values in the brackets for scenario V denote the other (possible) solution for sea ice parameters.

| Scenario | $\overline{hi}$ (m) | | | | $\overline{hs}$ (m) | | | |
|---|---|---|---|---|---|---|---|---|
| | Observed | Retrieval w/ flat snow cover | Retrieval w/ local $s$ | Retrieval w/ global $s$ | Observed | Retrieval w/ flat snow cover | Retrieval w/ local $s$ | Retrieval w/ global $s$ |
| I | 1.28 | – | 0.95 | 0.93 | 0.124 | – | 0.167 | 0.171 |
| II | 2.25 | 2.00 | 2.23 | 2.30 | 0.202 | 0.263 | 0.207 | 0.195 |
| III | 2.46 | – | 2.50 (1.69) | 2.45 (1.88) | 0.172 | – | 0.168 (0.293) | 0.175 (0.263) |
| IV | 3.01 | – | 3.25 | 2.38 | 0.321 | – | 0.285 | 0.419 |
| V | 4.13 | 3.88 | 4.09 | 4.11 | 0.308 | 0.350 | 0.313 | 0.310 |

**Table 3.** Uncertainty estimation for typical retrieval scenarios.

(a) Typical scenarios for uncertainty estimation

| Scenario | Ice type | $\overline{hi}$ (m) | $\overline{hs}$ (m) |
|----------|----------|------|------|
| S.I | FYI | 1.307 | 0.127 |
| S.II | FYI | 2.549 | 0.171 |
| S.III | MYI | 3.009 | 0.265 |
| S.IV | MYI | 4.736 | 0.348 |

(b) Results

| Scenario | Relative uncertainty | Perturbations | | | |
|----------|---------------------|------|------|------|------|
| | | *TB* | $FB_s$ | *s* | All |
| S.I | $\sigma_{hi}/hi$ | 9.44 % | 15.75 % | 8.19 % | 15.30 % |
| | $\sigma_{hs}/hs$ | 14.38 % | 25.08 % | 13.06 % | 24.21 % |
| S.II | $\sigma_{hi}/hi$ | 4.90 % | 4.36 % | 3.42 % | 5.60 % |
| | $\sigma_{hs}/hs$ | 11.33 % | 9.99 % | 7.89 % | 12.93 % |
| S.III | $\sigma_{hi}/hi$ | 11.15 % | 12.23 % | 5.02 % | 10.24 % |
| | $\sigma_{hs}/hs$ | 19.49 % | 21.41 % | 8.83 % | 17.88 % |
| S.IV | $\sigma_{hi}/hi$ | 6.62 % | 5.19 % | 4.71 % | 6.37 % |
| | $\sigma_{hs}/hs$ | 13.92 % | 10.94 % | 9.92 % | 13.28 % |

## A3 Radiative properties

The radiation model describes the radiation emitted from snow cover, sea ice and sea water, the brightness temperature (TB) at the top of atmospheric ($TB_{TOA}$) can be described as (Maaß et al., 2013b).

$$TB_{TOA} = (1-c)*(TB_{water} + (1 - e_{water})*TB_{cosm}) + $$
$$c*(TB_{ice} + (1 - e_{ice})*TB_{cosm}) + \Delta TB_{atm} \tag{A1}$$

5    In Equation A1, $c$ is sea ice concentration, $e_{ice}$ and $TB_{ice}$ the emissivity and TB of sea ice, $e_{water}$ and $TB_{water}$ are the emissivity and TB of sea water, $TB_{cosm}$ is cosmic microwave background radiation, which can be considered as uniform and constant (2.7 $K$). $\Delta TB_{atm}$ is TB from atmospheric contribution ranging from -0.36 $K$ and +5.67 $K$. Emissivity parameters are computed as follows: $e_{water}$ is based on the Fresnel equations in different directions of polarization (Ulaby et al., 1986) and $e_{ice}$ a function of parameters such as polarization, incidence angle, sea ice thickness, temperature, density, salinity, surface

10   roughness, snow depth and temperature. Based on Maaß et al. (2013b), the permittivity of snow ($\epsilon_{snow}$) is determined by a polynomial fit obtained from measurements at microwave frequencies ranging between 840 MHz and 12.6 $GHz$ (Tiuri et al., 1984) as follows.

$$\epsilon_{snow} = (1 + 0.7\rho_{snow} + 0.7\rho_{snow}^2) + i *$$
$$(1.59 \times 10^6 \times (0.52\rho_{snow} + 0.62\rho_{snow}^2) \times (f^{-1} + 1.23 \times 10^{-14}\sqrt{f})e^{0.036T})$$

Where $\rho_{snow}$ is the relative density of snow (compared to water), $T$ the temperature of snow in degrees Celsius and $f$ the microwave frequency. It should be noted that $\epsilon_{snow}$ depend on the snow wetness, which is not considered by the current model. Permittivity of sea ice ($\epsilon_{ice}$) is confirmed by brine volume fraction ($V_b$) using empirical relationship in Vant et al. (1978).

$\epsilon_{ice} = a_1 + a_2V_b + i * (a_3 + a_4V_b)$

$V_b$ is given in ‰, and the values of $a_1$, $a_2$, $a_3$, and $a_4$ follows Kaleschke et al. (2010). Similar to Maaß et al. (2013b), for the permittivity of sea water ($\epsilon_{water}$), empirical relationship from Klein and Swift (1977) is adopted, and the permittivity of air ($\epsilon_{air}$) is assumed to be 1. The brine volume fraction $V_b$ can be expressed in the following (Cox and Weeks, 1983).

$$V_b = \frac{\rho_{ice}S_{ice}}{\rho_{brine}S_{brine}(1+k)}$$

$S_{ice}$ is the ice salinity, $\rho_{ice}$ the ice density, and $S_{brine}$ the brine salinity and $\rho_{brine}$ the brine density. $\rho_{brine}$ can be fitted with $S_{brine}$ (in ‰) according to Cox and Weeks (1983).

$$\rho_{brine} = 1 + 0.0008 * S_{brine}$$

Then the following equation is adopted to relate $S_{brine}$ with $T_{ice}$ (Vant et al., 1978):

$$S_{brine} = a + b * T_{ice} + c * T_{ice}^2 + d * T_{ice}^3$$

Where $T_{ice}$ is in $°C$ and $a,b,c,d$ are fitted parameters in Vant et al. (1978). These polynomial approximations agreed well with the experimental data in Zubov (1963). Also, $\rho_{ice}$ can be expressed by ice temperature ($T_{ice}$: $°$C) in Pounder (1966):

$$\rho_{ice} = 0.917 - 1.403 \times 10^{-4}T_{ice}$$

Therefore, $V_b$ can be expressed as a function of $\rho_{ice}$, $S_{ice}$ and $T_{ice}$.

As derived model from Burke et al. (1979), the radiation model is a non-coherent model. However, the effect of non-
coherency is considered to be mitigated by several factors. First, since with the SMOS observations, there exists large variability of both sea ice thickness and snow depth within the typical resolution of 40 $km$. There usually exists large variability of the sea ice cover within the spatial scale of 40 $km$ (variability of $hi$ larger than 1/4 of L-band wavelength), which effectively

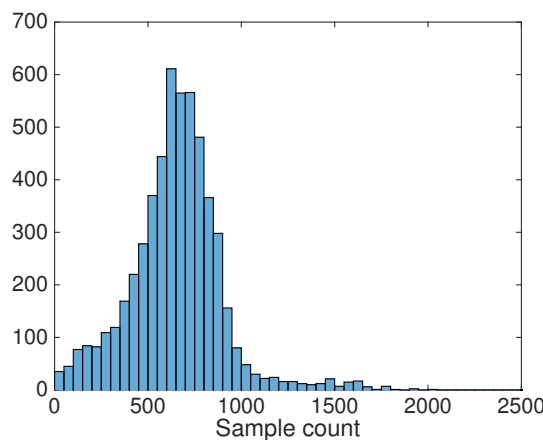

**Figure A1.** Distribution of OIB sample count ($M$).

mitigates the effect of non-coherency, as indicated in Kaleschke et al. (2010). Furthermore, multi-angle mean of SMOS TB further introduces a range of integration path of radiations. The multi-layer treatment of the sea ice also explored in Maaß et al. (2013b). According to the study in Zhou et al. (2017), with treatment of the salinity profile in MYI (i.e., salinity drainage in the top layers), the modeled TB is more consistent with the SMOS TB.

Under typical winter Arctic conditions ($T_{surf} = -30^\circ C$), simulated brightness temperature (TB) over different sea ice type from reformulated radiation model shows in Figure 3.c and d, with constant snow freeboard lines shown.

## A4    Radiation model verification with OIB and SMOS data

Verification is carried out between OIB data and SMOS, with specific attention to the effect of better OIB sampling. Due to the resolution difference between SMOS and OIB (or any type of satellite altimetry), we consider the case involving multiple ($M$)

OIB samples and a single SMOS TB. Section 2.2 and Figure 2 provide details of the correspondence between the two types of observations. Using all OIB data, Figure A1 shows the distribution of $M$ for all the retrieval problems (each corresponding to an area of 37.5 $km \times$ 37.5 $km$). The mean value of $M$ is about 700. The RMSE of TB (modeled v.s. observed by SMOS) is 3.1 $K$ for all available OIB data (see also Figure 3.c). If we further limit the computation of RMSE to the points with large values of $M$ (95-percentile for $M$, corresponding to areas with good OIB coverage), the RMSE drops to 1.41 $K$. Figure A2 shows the

relationship of TB error and $M$. As shown, there exists drop in both RMSE and the maximum error of TB with better spatial coverage of OIB. The lead information can be further incorporated in the radiation model (Zhou et al., 2017), which effectively reduces the overestimation of TB as caused by refrozen leads or open water.

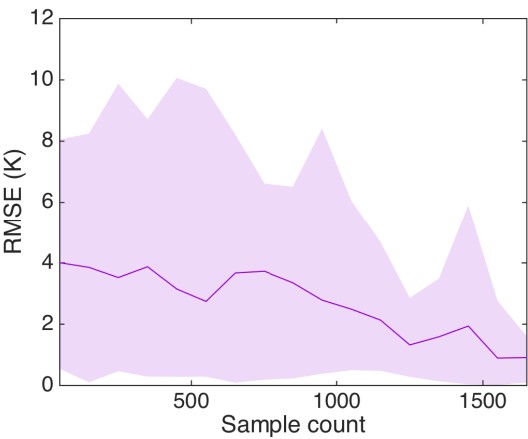

**Figure A2.** The relationship of RMSE of TB to OIB sample count ($M$). The statistics of RMSE of TB are computed for each sample count bin (each of 100). Shaded area covers the 5-th and 95-th percentile of the absolute TB error.

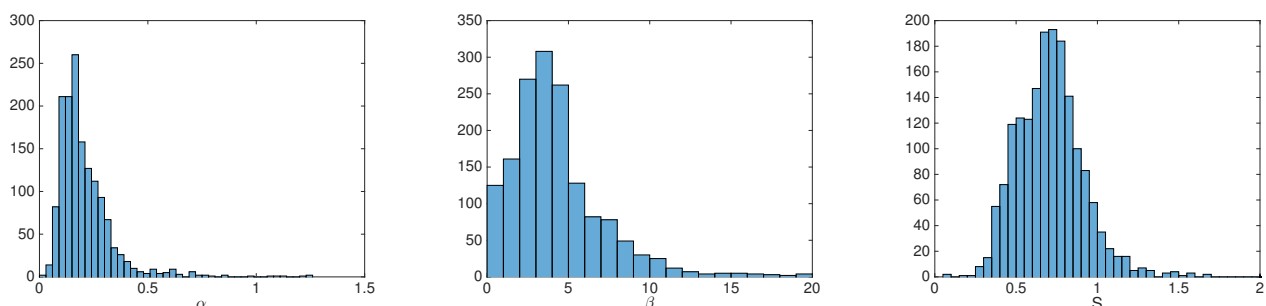

**Figure B1.** Distribution of $\alpha$, $\beta$ and $s$ for FYI. 40 $m$ resolution (OIB) data are used for computing the value of each parameter on the scale of 37.5 $km$ (i.e., approximately the native resolution of SMOS TB).

## Appendix B: Statistical analysis for covariability

We summarize the statistics of the fitted parameters of $\alpha$, $\beta$ and $s$ ($s = \alpha \cdot \beta$) as in Equation 3, based on all available OIB data. Figure B1 and B2 show the distribution of these parameters for FYI and MYI, respectively. The original resolution of OIB dataset (i.e., 40 $m$) is adopted. Further, the scaling of the distribution of $s$ is analyzed, by manually coarsening the OIB measurements by averaging adjacent samples. There exists statistically significant positive correlation between $hs$ and $FB_s$ across the scales from 40 $m$ to 240 $m$, and Figure B3 shows that the distribution of $s$ is stable across these scales, with a slight shift of its mode to a larger value at coarser scales. By cutting the distribution of $s$ at its 99-th percentile for FYI and MYI, we find that it can be well captured by Beta Distributions. Section 4.2 contains the quantified results of the fitted parameters for Beta Distributions at 40 $m$ resolution for both FYI and MYI.

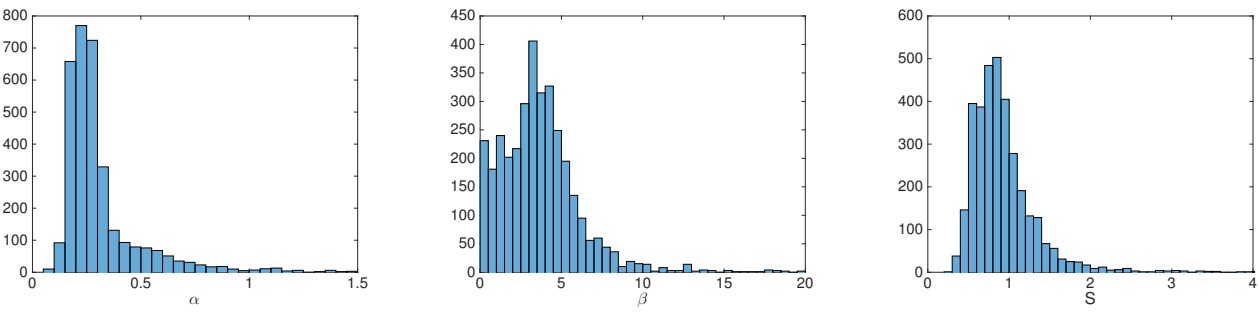

**Figure B2.** Same as Figure B1 but for MYI.

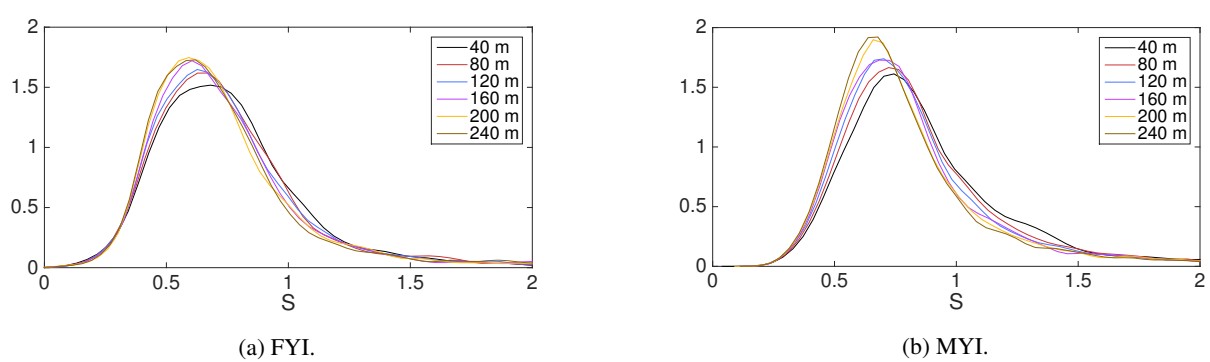

(a) FYI.                                                    (b) MYI.

**Figure B3.** Scaling of the distribution of $s$. Through coarsening the measurement of snow depth and snow freeboard with adjacent points, the value of $s$ is computed for each spatial scale with the nonlinear fitting in Section 3.1. The distribution of $s$ is computed using all available data from OIB tracks.

*Competing interests.* The authors declare no conflict of interest