# Peer review of "On the Retrieval of Sea Ice Thickness and Snow Depth using Concurrent Laser Altimetry and L-Band Remote Sensing Data"

_The Cryosphere, 2017_

## Referee Comment (RC1) · Anonymous Referee #1 · 24 Aug 2017

General comments:

In this manuscript the authors present a new method for the retrieval of sea ice thickness and snow depth using combined laser altimetry and L-band remote sensing data. The paper address a relevant scientific question within the scope of TC. The authors introduce a method based on snow freeboard derived from airborne Operation Ice Bridge (OIB) and the 1.4 GHz brightness temperature from the SMOS satellite. For verification they use again the OIB data and show a "good match" between observed and retrieved parameters.

The combination of altimetry and radiometry could possibly be used to improve re-

trievals of sea ice parameters. However, I am very sceptical that the conclusions derived in this manuscript are valid. The manuscript lacks a certain level of detail and traceability. Moreover it seems that the conclusions are based on circular reasoning. At least it seems that there is no real validation because both the retrieval and verification are based on the same data set. Therefore, I can not recommend to accept the paper for publication in The Cryosphere.

The authors use laser-based snow freeboard and radar-based snow thickness as well as the sea ice thickness from OIB flights. One problem of the present manuscript arises from the fact that the sea ice thickness in the OIB product was derived using the snow freeboard and the radar-based snow thickness (Kurtz et al., 2013). Therefore, it is not very surprising that the authors find a very high covariance between snow depth, thickness and freeboard because the quantities can not be considered as results from independent measurements. The "verification" in Figure 7 seems to just exemplify this circular reasoning. Additional problems arises from the fact that different snow depth algorithms for OIB exist and that the instrumentation has changed from year to year (Kwok et al., 2017). The dependency of the used data sets and the lack of discussion of the used assumptions cast serious doubts on the validity of the conclusions.

Regarding the use of L-band brightness temperatures I found the Figure 1 misleading with emissions arising just from the surface and not from the ice volume or deeper layers. Some crucial model assumptions are not explained in the manuscript, e.g. the parameterization of ice salinity. It seems that a thickness dependent parameterization was used otherwise I can not explain the large sensitivity to ice thickness exceeding 3 meter. All this should be explained and discussed in the manuscript. This is perhaps described in the reference Zhou et al. (accepted) but is not yet available to me.

Another issue with the manuscript is the overall aim of the method. It is not yet clear what is the main advantage of combining laser altimetry and L-band radiometry? Is the method for the fusion of airborne and satellite data or for to be used for future satellite missions? For the ICESat period there are no L-band radiometer data available. For

the ICESat-2 period it is not clear if SMOS and/or SMAP are still in operation. What about the different spatial and temporal samplings and uncertainties? These practical considerations are not yet even mentioned.

Specific comments:

P4L7 The resolution of the radiometer depends mainly on the size of the antenna.

Units should not be in italics.

---

## Author Comment (AC2) · 4 Sep 2017

The authors would like to make a further reply to the referee's comments to update a reference that has just been available online. This reference is Zhou et al. (2017), which describes the improved L-band radiation model for the sea ice. This model is applied in the retrieval algorithm which combines L-band brightness temperature (TB) observations and laser altimetry measurements. A multi-layer formulation is adopted in the model, with a sea ice type dependent salinity profile. The L-band TB saturates with the deepening of the sea ice depth (Figure 3 in the reference), which is qualitatively consistent with that in Maaß et al., (2013). We consider the model's performance is

guarantee for its usage in the retrieval algorithm. The referee is kindly directed to Section 4.1 and 4.2 of this reference for the validation with SMOS data. This reference is published online at: http://dx.doi.org/10.1080/01431161.2017.1371862.

The authors would also like to further emphasize that the saturation of TB with sea ice thickness does NOT pose a problem of sensitivity for the retrieval with TB and snow freeboard (measured by laser altimetry). TB saturates under a constant value of snow depth, as mentioned above, but the retrieval is carried out under a certain value of snow freeboard (NOT snow depth). There still exists good retrievability for both snow depth and sea ice thickness even if the sea ice is thick, as shown in Figure 3.c of the submitted manuscript.

References:

Maaß, N., Kaleschke, L., Tian-Kunze, X., and Drusch, M.: Snow thickness retrieval over thick Arctic sea ice using SMOS satellite data, The Cryosphere, 2013, 7, 1971–1989, doi:10.5194/tc-7-1971-2013..

Lu Zhou, Shiming Xu, Jiping Liu, Hui Lu and Bin Wang, Improving L-Band Radiation Model and Representation of Small-Scale Variability to Simulate Brightness Temperature of Sea Ice, International Journal of Remote Sensing, 38:23, 7070–7084, doi:10.1080/01431161.2017.1371862.

---

## Referee Comment (RC2) · Anonymous Referee #1 · 12 Sep 2017

Thank you for the rebuttal and providing the additional reference to the publication in the International Journal of Remote Sensing. The idea with the combination of lead data is very good. However, after reading the other paper even more questions arose because the publication also lacks traceability. I would not be able to reproduce your results with the somehow limited information given in the paper. Really important information is missing, some details are even wrong. For example, the used permittivity is not even mentioned. The native SSMI resolution of the 19 GHz channels used in the NASA-Team algorithm (Cavalieri et al. 1996) is about 69x43 not 10 km.

Regarding the present manuscript submitted to The Cryosphere, I am still not happy

with your answers. I still think you could be fooled by circular reasoning. You should split the data into independent "training" and "test" parts to do a real verification, and this for multi-year and first year ice separately. Thereby the very different resolutions and spatial samplings of the sensors involved have to be carefully considered.

---

## Referee Comment (RC3) · Anonymous Referee #2 · 15 Sep 2017

The manuscript "On the retrieval of sea ice thickness and snow depth using concurrent laser altimetry and L-Band remote sensing data" deals with the possibility to retrieve snow and ice thickness simultaneously from two different satellite missions. The goal is to combine brightness temperatures (TB) as measured for example by the SMOS L-band satellite and freeboard (FB) heights as measured for example by laser altimeter ICESat(-2). While the idea sounds very interesting and would be extremely beneficial to the polar science community, the presented approach is not convincing: The benefit of using a combination of L-band and altimeter measurements is not shown at all, and I doubt that the results from this work are representative for a potential "actual" retrieval from satellite data (see comments A). Furthermore, the authors have sometimes not been very careful with their citations (see comments B). The manuscript would also benefit from improving the readability and the language (see comments D, which is not a complete list). Further comments are given in C.

**Comments A:**
The major concerns I have about the presented study are stated here:

a) The authors do not discuss or show any results at all on the advantage of their suggested approach. If the new contribution here is to combine L-band and altimeter measurements, why do you not show the difference between using this combination of TB and freeboard measurements as compared to just using freeboard measurements and the relationship that you found between snow depth (hs) and FB? This could be done by 1) finding global alpha and beta values (instead of a global s=alpha*beta) and then using these alpha and beta values and Eq. 3 to convert FB to hs and then ice thickness (hi) (using Eq. 1 afterwards) or 2) finding a good fitting formulation that relates hs and hi and then converting FB to hs and hi using Eq. 1. The correlation between FB and hs ($R^2=0.67$) is very similar to the correlation found between the retrieved and the OIB-observed hs for using the global s values and SMOS TBs ($R^2=0.64$) / simulated TBs ($R^2=0.65$). Maybe it is essentially the correlation between FB and hs that is behind this agreement of retrieved and observed hs values? From the presented results, I cannot see whether there is any advantage in using L-band additionally...

b) The retrieval as performed in this study is not very representative for an "actual" retrieval:
1. Only 50% of the available SMOS grid cells are used for the analysis, based on the criterion that "the error" (do you mean RMSE here?) between simulated and SMOS-observed TB is < 1.5 K (as compared to 3.1 K for all SMOS cells). In a "real" retrieval situation where we do not have the information from OIB (i.e. hi, hs, and surface temperature) to simulate TBs, how can we identify these cases where simulated and SMOS-observed TBs differ more?
2. In the retrieval here, TBs are simulated taking the surface temperature from OIB measurements. In a "real" retrieval situation, one would have to use other sources for surface temperature, which have different spatial and temporal resolutions; they could be (and most likely are) measured with many hours time shift! And as surface temperature has been shown to have a huge influence on L-band TB (see e.g. Maaß et al., 2013) and surface temperature can vary on short time scales, the results can be very different when using other temperature information. This is shortly mentioned at the very end of the manuscript, but it is not stressed how much this can influence the retrieval performance.
3. The retrieval is done for freeboard measurements from the OIB campaign's laser altimeter. It is not discussed how the freeboard data from this altimeter compare with satellite based laser altimeters, which are given as the target for an "actual" retrieval.

c) With the current presentation of results I am not convinced that this is more than "playing around" with the relationship between ice thickness and snow depth (as observed during the OIB flights), which is taken advantage of. One would expect a relationship in that older ice is on average thicker and has had more time to accumulate snow on top, which leads to on average thicker snow on thicker ice (considering larger scales). And of course snow depth and freeboard are correlated as snow depth and ice thickness (together with snow and ice densities) determine the freeboard. Thus, I would not consider it as an "result" that you found a correlation between them. One interpretation (assuming that the densities of ice and water are relatively constant) is that the obtained $R^2$ value shows the fraction of the FB that is actually snow ($R^2=0.53$ would then mean that about 53% of the FB part consists of snow, while the rest is ice). This is also pointed out in the paper by Kwok et al. (2011), which is cited by the authors.

**Comments B:**
- Acknowledgements: OIB and SMOS data are not cited correctly here, see e.g. the NSIDC's and ICDC's (University of Hamburg) conditions/suggestions of how to cite the data usage

- p. 3, L7f: "Several recent studies focus on the retrieval of snow depth over thick sea ice, based on L-band passive microwave remote sensing data from SMOS (Tian-Kunze et al., 2012)." -> I am not aware of "several" studies, they are also not given here, and the given reference (Tian-Kunze et al., 2012) is not about retrieving snow depth but (thin) ice thickness.

- p. 4, L17: As far as I know, "the OIB Level-4 product IDCSI4" that is claimed to have been used for 2012 to 2015 in this study is only available for the years 2009 to 2013... (While the IDCSI2 Quicklook data is indeed available for 2012 to 2015)

- p. 4, L11-13: "...based on Burke et al. (1979). ... An adapted version of the model was adopted by Tian-Kunze et al. (2014) and ..." -> This is not correct. Tian-Kunze et al. (2014) use another (simpler) approach, which is originally based on a paper by Menashi et al. (1993). (also: "adapted version... was adopted" sounds strange)

**Comments C:**
- p. 2, L17: "schematic view of remote sensing of sea ice" -> this seems a bit exaggerated to me, the figure mainly shows the definitions of snow and ice thickness and freeboard...

- p. 2, L28-29: Not everyone knows what "its adapted version for ... FYI" is

- p. 3, L9: Here, "near realtime" observations are compared with altimetry "which can only achieve basin coverage on the scale of about one month" -> It would be helpful to add that SMOS provides not only "near realtime" data but also "an almost daily coverage of the polar regions".

- p. 3, L13: "Despite the limited coverage..." -> strange wording/argumentation

- p. 4, L7: "due to the limitation of satellite's orbital parameters, the inherent resolution is about 40 km" -> The resolution is determined by the antenna size, the frequency and the interferometry ("aperture synthesis") principle, not "orbital parameters".

- p. 4, L25-26: "we consider OIB measurements in the adjacent 3x3 cells ... of equal contribution" -> Maybe mention at least that this is an approximation (the contributions are actually not equal, see the SMOS "antenna gain function")

- p. 4, L30-31: "However, for certain segments of the OIB tracks, there exists extensive scanning which corresponds to a much larger value of M." -> I think it would be better and more precise to state the range of encountered M values (e.g. giving minimum, maximum and mean).

- p. 5, L3-5: "The purpose of these treatments is to rule out the factors that may compromise the quality of the OIB samples and allow focus on the discussion of the retrieval algorithm." -> However, these conditions that were excluded here do not only influence the OIB data quality but will probably also make a potential retrieval with SMOS data more difficult and should be discussed somewhere.

- p. 5, L16: Reference to used radiation model should be given already here.
From the given reference it is not clear how the authors of that paper have "reformulated the model to include multiple layers for sea ice and snow" (Zhou et al, 2017). If several ice layers are used in the model, the higher order reflection terms should be considered. I did not find a statement on this...

- p. 7, Eq. 3: Maybe better to use "arctan(...)" because tan^-1(...) could also be interpreted as 1/tan(...)

- p. 8, L5 - p. 9, L19: First you write about "scanning of alpha" without explaining it at all, then you present the results shown in Fig. 5. Then you explain the "scanning of alpha" procedure and finally refer to Tab. 2. This is very confusing. The scanning of alpha should be explained first. It would also be very interesting to see what values alpha takes in this procedure. Are they similar for the individual retrievals? Are they spread over a large range? Do the results shown in Fig. 5 and Tab. 2 belong to the same analysis?

- p. 10, L3-6:  "There is minor increase in quality (0.91 versus 0.89 and 0.65 versus 0.637) and a relatively large gap to the "ideal" case. This indicates that the uncertainty (or error) in TB and radiation models plays an important role in affecting the quality of the retrieval. The uncertainty of TB may arise from that of the radiation model, as well as the mismatch between the altimetry and passive microwave remote sensing"
I think the very similar results for simulated and SMOS TBs cannot lead to these conclusions about the radiation model or the TB measurements! Even if (theoretically) the radiation model gave completely unrealistic TB values: If you do the hs/hi-retrieval by comparing this radiation model's TBs (for different hs,hi values) with this same radiation model's "true" TB (for the "true" hs,hi values), the difference in retrieved and "true" hs,hi will originate from other assumptions used in the retrieval (here: assumption of Eq. 3, choice of s-value, choice of thresholds, ...) or from the ill-posedness of the problem (TB ambiguities in the model, which can also exist in reality) but not the quality of the model to represent the "real world"/SMOS (because you are comparing it with its own output! you are within its "ideal model world"). In contrast, an existing difference between using SMOS and simulated TBs may contain information on the radiation model's performance to simulate SMOS TBs (and also on the effect of the spatial mismatch between altimeter and satellite measurements).
As far as I can see, the difference between using global and local s values tells you something about how good the global s value approach is. Here (with the results for

simulated and SMOS TBs being very similar), THIS (=using different s values) is where the "relatively large gap to the 'ideal' case" seems to come from!

- Fig. 5 would benefit from a legend and/or annotations of some of the lines, it is hard to remember from the figure caption what each of the 7 lines represents...

**Comments D (not a complete list!):**
- Usage of "etc" is not very precise, for example on p. 2, L5 & p. 2, L20 & p. 4, L11 & p. 5, L34

- p. 2, L5-6: "there is rapid" -> "there has been rapid"

- p. 2, L24: "hence limited spatial coverage" -> not a complete sentence

- p. 3, L6: "researches...obtain snow depth" -> strange wording

- p. 3, L10: "requires the prerequisite" -> either "requires" or "prerequisite"

- p. 3, L25: "achieves successfully retrieval" is not a correct expression

- p. 3, L26: "correspond" -> "corresponds"

- p. 4, L20: "Temporally, the date of each OIB campaign is located, and the SMOS TB data from the specific date is attained for the combined retrieval." -> An example for a case where the readability could be improved. This sounds like a complicated way of saying something like: "OIB and SMOS data from the same day are taken."

- p. 4, L23-24: "due to the inherent resolution of SMOS data is about 40km, therefore..." is not a correct expression

- p. 4, L26: "the 9 cells covers" -> "the 9 cells cover"

- p. 4, L28: "the total area the contributes" -> "the total area that contributes"

- p. 6, L28 - p. 7, L7: This part is hard to understand...

- p. 8, L1-2: "For $h_s > 0$, there will be inundation due to: $FB < h_s$." -> Do you mean: "For $h_s > 0$, there will be inundation for values $h_s > FB$."?

---

## Author Comment (AC3) · 21 Oct 2017

The authors would like to thank the referee for the prompt and precise comments to our reply. We have made modifications and the accompanying reply as follows.

**The referee's comments:**

Thank you for the rebuttal and providing the additional reference to the publication in the International Journal of Remote Sensing. The idea with the combination of lead data is very good. However, after reading the other paper even more questions arose because the publication also lacks traceability. I would not be able to reproduce your results with the somehow limited information given in the paper. Really important information is missing, some details are even wrong. For example, the used permittivity is not even mentioned. The native SSMI resolution of the 19 GHz channels used in the NASA-Team algorithm (Cavalieri et al. 1996) is about 69x43 not 10 km.

**Reply:**

With respect to the comment that key information of the L-band radiation model is not available in Zhou et al. (2017), we consider it necessary to formulate a supplementary to describe the model in detail. Specifically, this supplementary includes introduction to the model, including the multi-layer treatment of salinity and temperature profile, sea ice emissivity and permittivity, etc. The modeled L-band brightness temperature (TB) to a range of sea ice parameters for firstyear ice (FYI) is also included (Figure S1.a of the supplement), accompanying that for multiyear ice (MYI) in Figure 3c in the manuscript (also Figure S1.b of the supplementary). This supplementary is provided as a supportive document to the original manuscript, and also attached at the end of this reply document.

The following is a concise answer to the referee's question on the permittivity settings in the model: the permittivity of snow, sea ice and sea water mainly follows that in Maaß, et al., (2013). For sea water, an empirical relationship in Klein and Swift (1977) is adopted assuming salinity of 33 g/kg. For sea ice, the permittivity is related to the brine volume fraction, based on Vant et al. (1978) and specific settings in Kaleschke et al. (2010). For snow, same as Maaß, et al. (2013), we adopt fitted parameterization of permittivity as formulated in Tiuri et al., (1984). For the study in this work, under a multi-layer formulation, the salinity structure of MYI (in terms of salinity for each layer) and its effect on permittivity is characterized. The attached supplement provides details of these model settings.

With respect to the comment that a wrong detail of the native SSMI resolution is present in Zhou et al. (2017), we thank the referee and recognize that this is indeed a mistake. We are preparing a corrigendum to the International Journal of Remote Sensing to correct the description, as indicated by the referee: *"Furthermore, sea ice concentration data as used by this article (Cavalieri et al. 1996) are provided on the resolution of 25 km with the native resolution of about 69 km by 43 km*". For a further reply: other datasets, such as the sea ice concentration based on 89 GHz channel of AMSR-E or AMSR-2 (see Spreen et al., (2008)) serves as candidates for sea ice coverage, which is of about 5 km in nominal resolution. The intention of using sea ice concentration (SIC) in Zhou et al. (2017) is to complement the lead maps in characterizing the effect of (refrozen) open water on the L-band TB of the sea ice cover. Sea ice leads are usually with much small width than the typical passive remote sensing with satellites, and not well represented in the retrieval algorithms for SIC. Therefore, both sea ice lead information and sea ice concentration information are adopted in simulating the TB (see Equation 5 and the last paragraph on page 7075 of Zhou et al., (2017)).

**The referee's comments:**

Regarding the present manuscript submitted to The Cryosphere, I am still not happy with your answers. I still think you could be fooled by circular reasoning. You should split the data into independent "training" and "test" parts to do a real verification, and this for multi-year and first year ice separately. Thereby the very different resolutions and spatial samplings of the sensors involved have to be carefully considered.

The authors would like to make further clarifications on the issue of circular reasoning, and according to the suggestions of the referee, we carry out experiments with different portion of the dataset and show the results.

First, we would like to clarify that the retrieval is based on the snow freeboard (from OIB) and L-band TB (from SMOS), with L-band radiation model and hydrostatic relationship. The L-band model is not trained to the observations of OIB or SMOS. Besides, the model is qualitatively consistent with existing works such as Maaß et al., (2013). The other supportive information for the retrieval is the covariability between snow depth (*Hs*) and snow freeboard (*FB*snow), which is based on OIB data on the spatial scale of 40 meters. However, it is important to note that the covariability in the retrieval is only a generic function shape between *FB*snow and *Hs* (Equation 3 of the manuscript), and this information does NOT directly specify *Hs* for any specific *FB*snow. Instead, under the constraints of both observations (*TB* and *FB*snow), the retrieval algorithm seeks the proper value of  $\alpha$  that directly decides *Hs* and *Hi*. Given typical distribution of *FB*snow for FYI and MYI (in Figure 1.a below), the scanning of  $\alpha$  and the resulting mean snow depth are shown in Figure 1.b. The parameter *s* that describes the aforementioned covariability (i.e., the function shape parameter) is set to 0.71 and 0.95 for FYI and MYI respectively, which is also adopted in the verification of the original manuscript.

Figure 1. Scanning of parameter  $\alpha$  and the corresponding mean snow depth.

With respect to the difference in the spatial sampling from various sensors as pointed out by the referee, the authors make the following clarifications on how this difference is accounted for during the retrieval, and analyze the relationship to existing satellite laser altimetry. For the retrieval, we assume that the laser altimetry returns the mean value of  $FB_{snow}$  on a certain spatial scale. For example, for ICESat, the spatial scale of each laser scan is about 70 meters, which is different from 40 meters of OIB (L4 data). It is worth noting that covariability between Hs and  $FB_{snow}$  is present on various spatial scales (Kwok et al., (2011)), and there is indeed scaling of the covariability across different scales (i.e., at different resolution for altimetry). Figure 2 (below)

shows the distribution of s derived from all OIB data on various scales, and there is a slight drift of s to smaller values when manually coarsening the altimetry measurements. This drift applies to both FYI and MYI. Furthermore, in Figure 3 (below), it is shown that there exists distinctive spatial distribution of s for MYI, with the regions north of the Canadian Archipelago featuring s>1. Since for the actual retrieval, the local value of s is not available, in the verification of the original manuscript (Figure 7c and d and corresponding parts), only the global mean value of s is adopted.

(a) FYI. (b) MYI. Figure 2. Scaling of *s* derived from OIB data.

(a) FYI. (b) MYI. Figure 3. Spatial distribution of *s*. 40 meter resolution (original OIB) is adopted.

With respect to the suggestions that two separate datasets for "training" and "testing" respectively to ensure independency, the authors have carried out experiments accordingly for further validation of the retrieval. First, the authors would like to clarify that, according to the understanding of the authors, the "training" process mentioned by the referee is the derivation of the value of *s*. Therefore, we have split all OIB data into years, and use all the data in year 2012 to 2014 to derive *s* for FYI and for MYI. This process is denoted "training", and the derived values of *s* are 0.73 and 1.00 for FYI and MYI, respectively. These values are applied to the retrieval with OIB data in year 2015 (with *FB*snow from OIB and SMOS TB), which is denoted the process of "testing". The  $R^2$  of the fitting between the retrieved *Hi* (*Hs*) to the observed *Hi* (*Hs*) is 0.88 (0.62) for linear fitting and 0.87 (0.61) for linear fitting line with the constraint that the slope be 1, respectively. The results are also shown in Figure 4 (below), which closely resembles those for the large-scale retrieval in the original manuscript (Figure 7c and d). This provides further verification on the consistency of the nonlinear fitting (in terms of *s*), and that both *Hi* and *Hs* can be retrieved

with the proposed method.

---

## Author Response (AR1)

**Summary to the reply and manuscript revision**

The authors would like to sincerely thank the two anonymous referees for the review of the original manuscript, and the discussion involved in the process. We consider these suggestions and comments invaluable for improving the manuscript's relevance and quality. There is a common question by both referees on the potential of circular reasoning and the independency of the data during verifications. In this summary, we would like to address this common question. For other specific comments, the replies are provided by compilation of reply documents to all the available commentary documents (RC1 to RC3).

Concerning the common question of the potential of circular reasoning and the independency of the data, we would like to provide the following proof. **First**, without the covariability, the retrieval can still be carried out by using mean $FB_{snow}$ and L-band TB, and with the covariability info that supports multiple $FB_{snow}$ samples, the quality of retrieval is improved for both $hi$ and $hs$. This shows that covariability ONLY plays a central role in general retrieval problem which involves high-resolution altimetry samples and (relatively) low resolution L-band radiometry. The evidence is now added in the revised manuscript (pg. 10, l. 31 to pg. 11, l. 6).

**Second**, without TB, the "retrieval" can be achieved with ONLY the covariability (by using the mean values for $alpha$ and $beta$), BUT the quality of retrieval is much compromised, especially for $hs$. This indicates that TB plays an important role in the combined retrieval process. The quantitative results are provided in the foremost part of the reply to RC3.

**Third**, with respect to the data dependency in the derivation of $s$ (the parameter that characterize covariability), we divide the OIB data by years. According to the suggestion from referee #1, the values of $s$ as derived from years before 2015 are used for the retrieval for OIB data in 2015 (to ensure independency). The results (shown in reply document to RC2) show that the retrieval can be carried out for data in 2015, with comparable performance for both $hi$ and $hs$.

**Fourth**, the accurately modeled TB ensures good retrieval, and whether the sea ice cover is sufficiently surveyed by altimetry scans plays an important role in the TB error (RMSE of TB dropping from 3.1 K for all data to 1.4 K for well-scanned areas). Quantitative result of TB error is provided in the supplementary material (as well as reply to RC3), and the effects on retrieval in the revised manuscript (pg. 10, l. 20 to l. 31).

The specific replies and the revised manuscript are provided after this summary. The contents of the following parts are outlined as followings. Within each part of the content, the figures and references are numbered and provided independently. The highlighted parts in the revised manuscript differs by color: the revisions according to the first referee's comments (RC1 and RC2) are in yellow, and those to the second referee's (RC3) are in green.

The authors sincerely hope that the contents of the manuscript are fully and accurately conveyed, and would like very much to reply any further potential questions from the referees and the editor.

**Contents:**

**Reply to RC1:**

The authors thank the referee #1 for the review and suggestion to our manuscript. We would like to make the following clarifications and responses to the comments.

*Comment from the referee:*
*The authors use laser-based snow freeboard and radar-based snow thickness as well as the sea ice thickness from OIB flights. One problem of the present manuscript arises from the fact that the sea ice thickness in the OIB product was derived using the snow freeboard and the radar-based snow thickness (Kurtz et al., 2013). Therefore, it is not very surprising that the authors find a very high covariance between snow depth, thickness and freeboard because the quantities can not be considered as results from independent measurements. The "verification" in Figure 7 seems to just exemplify this circular reasoning. Additional problems arises from the fact that different snow depth algorithms for OIB exist and that the instrumentation has changed from year to year (Kwok et al., 2017). The dependency of the used data sets and the lack of discussion of the used assumptions cast serious doubts on the validity of the conclusions.*

Concerning the judgment by the referee that the verification in the manuscript is "circular reasoning", we would like to point out that the verification is carried out based on the data independent from what used in the retrieval process. **Three clarifications are provided as follows.**

1. **Data independency in the covariability analysis.** Although the OIB data is used in the analysis, the covariability is carried out between two independently measured parameters: **the total freeboard** measured by airborne topographic mapper (ATM), and **snow depth** measured by ultra-wideband frequency-modulated continuous-wave (FMCW) radar (see Farrell et al., 2012 and related references). Thus the two observations can be considered as independent. Moreover, the covariability analysis is **NOT** carried out between the freeboard (or snow depth) and sea ice thickness, since sea ice thickness is a derived parameter based on the hydrostatic equilibrium relationship. Note that sea ice thickness of OIB is only used in the verification of the retrieved parameters.

2. **Covariability between snow depth and total freeboard is physically sound, and NOT due to that they are not independent.** Considering Equation 1 of the manuscript (isostatic equilibrium model) which relates snow depth, sea ice thickness and the total freeboard, the relationship can be expressed as follows (using the notation in the manuscript):

$$FB_s = \frac{\rho_w - \rho_i}{\rho_w} \cdot h_i + \frac{\rho_w - \rho_s}{\rho_w} \cdot h_s \approx 0.1 \cdot h_i + 0.67 \cdot h_s$$

As indicated by this relationship, if the value of snow depth is considered a random variable, the value of the total freeboard (also a random variable) is clearly correlated with it. Therefore, this covariability is inherent in the physics, NOT due to the data collection process (or OIB in this case). See also Kwok et al., (2011) (subfigure d of figure 10, 11 and 12) for a similar analysis which also provides justification of covariability, but on a different spatial scale of 4 km.

3. **Role of the covariability in the retrieval.** The covariability (between snow depth and total freeboard) is a statistical relationship derived from the OIB data, and the parameters derived from functional fitting (see Section 3.1) using basin-wide observations that are integrated into the retrieval. During the retrieval, the input data include the total freeboard and the L-band TB, and the output data are sea ice thickness and snow depth, while the covariability (in terms of the fitted parameters) serves as a supportive info. The only place where the local information of covariability is used is in the analysis of the sources of the

retrieval error in Section 3.2 and Section 4. During the actual retrieval, the local covariability information is NOT available, and Figure 7c and d shows that the corresponding retrieval results are in good consistency with the observed sea ice parameters. Therefore, to summarize the proofs, for the design of the retrieval algorithms and the verification, we does not consider there exists circular reasoning.

Second, concerning the quality of the OIB data especially the snow depth algorithms applied for each year, we make the following explanations of the details of OIB data used in the analysis. In Kwok et al. (2017) (also the reference provided by the referee), five retrieval algorithms (NISDC, GSFC-NK, SLRD, Wavelet, JPL) for snow depth in OIB are discussed. In these five algorithms, both NISDC and GSFC-NK retrievals have lower scatter compared to in situ campaigns and large inter-annual variability compared to the climatological fields. Kurtz et al., (2013a) indicates that IDSI4 product are capable of providing a reliable record of snow depth through independent data comparison. For our analysis and validation, we use IDSI4 dataset in 2009-2013 (Kurtz et al., 2015), which is the existing NSIDC product, and quick-look dataset in 2014–2016, which is based on GSFC-NK algorithm. Data from the quick-look data is used for the campaign which is not accessible since 2014. Quick-look product, which uses modified algorithms to minimize freeboard biases (Kurtz, 2013; Kurtz et al., 2014), takes the consideration of different instrument characteristics of the snow radar. Despite that it is possible that the uncertainties in this product are higher than in IDSI4 product (Kurtz et al., 2013b), we consider the data product as used in the manuscript are optimum in terms of the overall small bias and good consistency with in-situ measurements, for the purpose of the analysis of covariability and the verification of the retrieval algorithm.

Third, we want to emphasize that the main purpose of the manuscript is the introduction of the retrieval methodology, and the demonstration of its validity through verification with OIB data. Since the "true" values of both snow depth and sea ice thickness are available from OIB, we use the total freeboard measurements (to simulate satellite laser altimetry) and SMOS TB data for the retrieval and verification with the aforementioned "true" or reference values of OIB. Similar practice of using OIB data for verifications to the retrieval algorithm is also common. Take Maaß et al. (2013) as an example. The snow depth from OIB is used to verify the retrieved snow depth (figure 9 and 10), while the sea ice thickness from OIB is used to guarantee the prerequisite of the retrieval algorithm that the sea ice is thick enough. With respect to the data production with actual satellite data, the covariability that is specific to the resolution of the satellite altimetry should be used (which can be potentially derived from OIB data due to the higher resolution) and compared against other independent data sources for verification, such as other campaigns. We consider this an important direction of future work, which is beyond the length and scope of current work.

*Comment from the referee*:

*Regarding the use of L-band brightness temperatures I found the Figure 1 misleading with emissions arising just from the surface and not from the ice volume or deeper layers. Some crucial model assumptions are not explained in the manuscript, e.g. the parameterization of ice salinity. It seems that a thickness dependent parameterization was used otherwise I can not explain the large sensitivity to ice thickness exceeding 3 meter. All this should be explained and discussed in the manuscript. This is perhaps described in the reference Zhou et al. (accepted) but is not yet available to me.*

First, concerning the misleading plot (Figure 1), our original intention is to contrast the altimetry and passive remote sensing techniques. Here we modified the figure to avoid the unnecessary misleading info. The modified figure is shown below.

[Figure]

Modified figure 1. Schematic view of sea ice parameters and active/passive remote sensing of the sea ice cover. The parameters include sea ice thickness (hi), snow depth (hs) and snow freeboard (FBsnow).

Second, concerning the L-band radiation model used for the retrieval, the model is a multi-layer radiation model based on sea ice type-dependent salinity and temperature profile. The model is verified using OIB and SMOS observational datasets. The manuscript documenting the radiation model has been accepted by International Journal of Remote Sensing (Zhou et al., 2017), but it is not available for public at this moment. Here we would like to quote relevant description of the model and the accompanying figure as below. Please also see Section 2.3 and Figure 3.a (salinity profile) for other supportive information.

> *"In order to explore the sensitivity of the L-band radiation model to the properties of sea ice, we reformulate the model to include multiple layers for sea ice and snow, instead of a single layer adopted by Maass et al. (2013). Specifically, we use a linear vertical temperature prole for the sea ice and snow, based on prescribed thermal conductivity of sea ice and snow, following Untersteiner (1964) and Yu and Rothrock (1996).*
>
> *…⁻*
>
> *Furthermore, we distinguish the difference in salinity between first-year ice (FYI) and multi-year ice (MYI). For FYI, following Cox and Weeks (1974), a constant salinity prole based on the ice thickness is used to characterize the fact that the salinity has not drained. For MYI, a salinity prole based on observations is used to reflect the effect of brine drainage and flushing during the melt season.*
>
> *…*
>
> *We denote the original model as the single layer radiation model as adopted in Maass et al. (2013), and the improved model as the multi-layer model with ice type dependent vertical temperature and salinity prole. Figure 4.a shows the scatterplot between the original modeled and SMOS observed $T_B$ for 22 Mar 2012, while Figure 4.b is for the improved model. The original modeled $T_B$ tends to cluster around 250K, irrespective of the large range of observed $T_B$ values (y-axis). Clearly, the improved model produces a much better t than that of the original model (0.84 as compared to 0.60 for $R^2$). The overestimated $T_B$ by the original model for both FYI (triangles) and MYI (circles) are significantly reduced. Figure 4.c (4.d) show the scatterplot for 18 Mar 2014 and 1 Apr 2015 under the original model (the improved model). Again, the simulated $T_B$ is in much better agreement with observations (0.01 as compared to 0.68 for $R^2$). Taking a close look at Figure 4.b and 4.d, we note that the simulated $T_B$ on 18 Mar 2014 and 1 Apr 2015 is not tightly clustered along the t line as that of 22 Mar 2012, which is also reflected by the relatively lower $R^2$ in Figure 4.d as compared to Figure 4.b. For these two days, there exist extensive leads as observed by OIB campaigns, which are small in width and not directly distinguishable by L-band observations of SMOS.*

...”

**Figure 4.** Comparison of modeled and observed $T_B$ for OIB data from representative days. Subfigure $a$ to $d$ compares the original model and the improved model (with the multi-layer formulation and the vertical salinity and temperature profile). Subfigure $a$ (or $c$) and $b$ (or $d$) show the evaluation of the original model and the improved model for 22 Mar 2012 with no presence of sea ice leads on the OIB track (or 18 Mar 2014 and 1 Apr 2015 with the OIB tracks covering sea ice leads). The effect of the integration of MODIS lead map is further compared by subfigure $e$ (with the improved model but without lead information) and $f$ (with lead information). FYI and MYI are marked out by triangles and circles, respectively. Red lines are the least square fit, whilst blue lines are the least square under the constraint that the slope of the fit line is 1. Black lines are 1:1 lines.

We would also like to provide the accepted manuscript (Zhou et al., 2017) to the referee upon request. Furthermore, we want to emphasize the sensitivity for sea ice over 3 meters thick arises from that the retrieval of ice thickness is based on the total freeboard and L-band TB. The referee is kindly directed to Figure 3c, in which the relationship between the L-band TB and sea ice parameters is shown. The retrieval is carried out under a certain total freeboard value (shown by green constant-freeboard lines in the figure), but a certain snow depth. Therefore, even the TB saturates when sea ice is thick under a prescribed snow depth (black lines in Figure 3c), for the proposed algorithm there still exists good retrievability. Section 2.3 (line 26 on page 5) gives a more thorough description on this issue.

***Comment from the referee***:

*Another issue with the manuscript is the overall aim of the method. It is not yet clear what is the main advantage of combining laser altimetry and L-band radiometry? Is the method for the fusion of airborne and satellite data or for to be used for future satellite missions? For the ICESat period there are no L-band radiometer data available. For the ICESat-2 period it is not clear if SMOS and/or SMAP are still in operation. What about the different spatial and temporal samplings and uncertainties? These practical considerations are not yet even mentioned.*

First, concerning the advantage of the proposed retrieval algorithm, we would like to further emphasize the status-quo of the retrieval of snow depth and sea ice thickness (see also Section 1, the second and the third paragraph). Current retrieval algorithms mainly focus on a single type of sea ice parameter (such as ice thickness or snow depth), and thus exists large uncertainty due to the lack of knowledge of other parameters. The aim of the proposed algorithm is to retrieve both parameters simultaneously, without simple assumptions such as the snow depth estimations (from climatology or reanalyses) in laser altimetry. The retrieved parameters should be able to serve as better estimations of these parameters and serve potential climatological and operational usage.

Second, concerning the retrieval with simultaneous satellite campaigns, we mainly target at the synergy of observational data between ICESat-2 and SMOS/SMAP. The manuscript provides a basis for the retrieval algorithm design with actual satellite data. ICESat-2 is currently scheduled for launch in 2018

(see https://icesat-2.gsfc.nasa.gov). SMOS and SMAP have been in service since late 2009 and early 2015. Although the designed lifespan of SMOS and SMAP are both 3 years, it is worth noting that SMOS have been providing service for over 7 years. Since it is invaluable for the availability of satellites that co-register the interested regions with complementary capabilities, we would like to express their optimism in the satellite campaigns and determination to make better usage of potential data for the retrieval of sea ice parameters. Additionally, the ongoing Chinese satellite campaign WCOM (Shi et al., 2016) with passive microwave remote sensing (including L-band) capabilities will cover the Arctic region, which serves as another candidate dataset. WCOM is scheduled to launch before 2020. For the co-registration of WCOM and ICESat-2, the combined retrieval can be carried out for the corresponding total freeboard and L-band TB measurements.

***Comment from the referee***:
*P4L7 The resolution of the radiometer depends mainly on the size of the antenna.*

According to the comment, the authors have made the correction to the manuscript with respect to the resolution of radiometry.

***The referee's comments:***

*Regarding the present manuscript submitted to The Cryosphere, I am still not happy with your answers. I still think you could be fooled by circular reasoning. You should split the data into independent "training" and "test" parts to do a real verification, and this for multi-year and first year ice separately. Thereby the very different resolutions and spatial samplings of the sensors involved have to be carefully considered.*

The authors would like to make further clarifications on the issue of circular reasoning, and according to the suggestions of the referee, we carry out experiments with different portion of the dataset and show the results.

First, we would like to clarify that the retrieval is based on the snow freeboard (from OIB) and L-band TB (from SMOS), with L-band radiation model and hydrostatic relationship. The L-band model is not trained to the observations of OIB or SMOS. Besides, the model is qualitatively consistent with existing works such as Maaß et al., (2013). The other supportive information for the retrieval is the covariability between snow depth ($Hs$) and snow freeboard ($FB_{snow}$), which is based on OIB data on the spatial scale of 40 meters. However, it is important to note that the covariability in the retrieval is only a generic function shape between $FB_{snow}$ and $Hs$ (Equation 3 of the manuscript), and this information does NOT directly specify $Hs$ for any specific $FB_{snow}$. Instead, under the constraints of both observations ($TB$ and $FB_{snow}$), the retrieval algorithm seeks the proper value of $\alpha$ that directly decides $Hs$ and $Hi$. Given typical distribution of $FB_{snow}$ for FYI and MYI (in Figure 1.a below), the scanning of $\alpha$ and the resulting mean snow depth are shown in Figure 1.b. The parameter $s$ that describes the aforementioned covariability (i.e., the function shape parameter) is set to 0.71 and 0.95 for FYI and MYI respectively, which is also adopted in the verification of the original manuscript.

[Figure]

(a) $FB_{snow}$ distribution.        (b) Mean snow depth.

Figure 1. Scanning of parameter $\alpha$ and the corresponding mean snow depth.

With respect to the difference in the spatial sampling from various sensors as pointed out by the referee, the authors make the following clarifications on how this difference is accounted for during the retrieval, and analyze the relationship to existing satellite laser altimetry. For the retrieval, we assume that the laser altimetry returns the mean value of $FB_{snow}$ on a certain spatial scale. For example, for ICESat, the spatial scale of each laser scan is about 70 meters, which is different from 40 meters of OIB (L4 data). It is worth noting that covariability between $Hs$ and $FB_{snow}$ is present on various spatial scales (Kwok et al., (2011)), and there is indeed scaling of the covariability across different scales (i.e., at different resolution for altimetry). Figure 2 (below) shows the distribution of $s$ derived from all OIB data on various scales, and there is a slight drift of $s$ to smaller values when manually coarsening the altimetry measurements. This drift applies to both FYI and MYI. Furthermore, in Figure 3 (below), it is shown that there exists distinctive spatial distribution of $s$ for MYI, with the regions north of the Canadian Archipelago featuring

*s>1*. Since for the actual retrieval, the local value of *s* is not available, in the verification of the original manuscript (Figure 7c and d and corresponding parts), only the global mean value of *s* is adopted.

[Figure]

(a) FYI.          (b) MYI.

Figure 2. Scaling of *s* derived from OIB data.

[Figure]

(a) FYI.          (b) MYI.

Figure 3. Spatial distribution of *s*. 40 meter resolution (original OIB) is adopted.

With respect to the suggestions that two separate datasets for "training" and "testing" respectively to ensure independency, the authors have carried out experiments accordingly for further validation of the retrieval. First, the authors would like to clarify that, according to the understanding of the authors, the "training" process mentioned by the referee is the derivation of the value of *s*. Therefore, we have split all OIB data into years, and use all the data in year 2012 to 2014 to derive *s* for FYI and for MYI. This process is denoted "training", and the derived values of *s* are 0.73 and 1.00 for FYI and MYI, respectively. These values are applied to the retrieval with OIB data in year 2015 (with $FB_{snow}$ from OIB and SMOS TB), which is denoted the process of "testing". The $R^2$ of the fitting between the retrieved *Hi* (*Hs*) to the observed *Hi* (*Hs*) is 0.88 (0.62) for linear fitting and 0.87 (0.61) for linear fitting line with the constraint that the slope be 1, respectively. The results are also shown in Figure 4 (below), which closely resembles those for the large-scale retrieval in the original manuscript (Figure 7c and d). This provides further verification on the consistency of the nonlinear fitting (in terms of *s*), and that both *Hi* and *Hs* can be retrieved with the proposed method.

[Figure]

(a) Sea ice thickness.  (b) Snow depth.

Figure 4. Verification of retrieval in year 2015. OIB data between 2012 and 2014 are used for the derivation of the values of *s*. The solid line is the 1:1 line and the dashed (dash-dotted) line represents the linear fitting (linear fitting line with the constraint that the slope be 1)

The authors thank the referee, and consider the referee's comments invaluable in making the manuscript clearer and more relevant. We hope that through the responses above, we can fully convey our idea on the retrieval, and would like to answer potential questions from the referee for further clarification. The materials above (including the supplementary material of the radiation model) are considered by the authors to be merged into the original manuscript in its revised form.

**Reply to RC3:**

The authors would like to thank the referee for the insightful review of the manuscript and the suggestions for the revision of the manuscript. Replies to the comments by the referee are as follows. The original comments from the referee are shown in italic Arial font, and the reply to each comment is provided immediately after the comment and is in Times font.

*Comments A:*

*The major concerns I have about the presented study are stated here:*

*a) The authors do not discuss or show any results at all on the advantage of their suggested approach. If the new contribution here is to combine L-band and altimeter measurements, why do you not show the difference between using this combination of TB and freeboard measurements as compared to just using freeboard measurements and the relationship that you found between snow depth (hs) and FB? This could be done by 1) finding global alpha and beta values (instead of a global s=alpha\*beta) and then using these alpha and beta values and Eq. 3 to convert FB to hs and then ice thickness (hi) (using Eq. 1 afterwards) or 2) finding a good fitting formulation that relates hs and hi and then converting FB to hs and hi using Eq. 1. The correlation between FB and hs (R^2=0.67) is very similar to the correlation found between the retrieved and the OIB observed hs for using the global s values and SMOS TBs (R^2=0.64) / simulated TBs (R^2=0.65). Maybe it is essentially the correlation between FB and hs that is behind this agreement of retrieved and observed hs values? From the presented results, I cannot see whether there is any advantage in using L-band additionally...*

The authors would like to emphasize that the main advantage is the simultaneous retrieval of BOTH *Hi* and *Hs*, by combining L-band passive remote sensing and laser altimetry. The covariability information that is incorporated into the retrieval does NOT specify *Hs*. Rather, this information is only a general constraint between $FB_{snow}$ and *Hs*. According to the suggestions from the referee, we further carried out "retrieval" study using only the statistics derived from OIB, and NOT using TB. Specifically, the statistics are computed for the parameters *alpha*, *beta*, and *s*. Results are shown in Figure 1 and 2 (below) for FYI and MYI, respectively. Figure 3 (below) shows the retrieval results for *Hi* and *Hs* using the globally fitted *alpha* and *beta* on the exactly same spatial scale of the retrieval in the manuscript. The adopted values for these parameters are specific to FYI and MYI: for FYI, *alpha*=0.21 and *beta*=3.38; for MYI, *alpha*=0.31 and *beta*=3.07.

[Figure]

(a) *alpha*.       (b) *beta*.       (c) *s*.

Figure 1. Distribution of alpha, beta and s for FYI. 40-meter resolution (OIB) data
are used for computing the value of each parameter on the scale of 37.5 km
(i.e., approximately SMOS TB resolution)

[Figure]

(a) *alpha*.        (b) *beta*.        (c) *s*.

Figure 2. Distribution of alpha, beta and s for MYI. Specifications are the same as Figure 1.

[Figure]

(a) *Hi*.          (b) *Hs*.

Figure 3. Retrieval of *Hi* and *Hs* using globally fitted *alpha* and *beta*.

As shown in Figure 3, with the covariability information (*alpha* and *beta*), indeed retrieval can be carried out. The least squares fit (dashed lines) for FYI (blue) and MYI (red) for *Hi* yields 0.93 and 0.96 respectively for $R^2$, and that for *Hs* yields 0.51 and 0.77. However, the least squares fit (dotted-dashed lines) for *Hs* on FYI (blue) yields negative $R^2$, and that for *Hs* on MYI (red) is as low as 0.15. For example, for MYI, there exists large overestimation of *Hs* when *Hs* is small, and underestimation of *Hs* when it is large.

Table 1. $R^2$ for the least-squares fit (slope=1) for the retrieval.

| | Retrieval with $FB_{snow}$, SMOS TB, and global *s* | Retrieval with $FB_{snow}$ and covariability (*alpha* and *beta*) |
|---|---|---|
| *Hi* in FYI | 0.96 | 0.93 |
| *Hi* for MYI | 0.83 | 0.96 |
| *Hs* for FYI | 0.78 | -- |
| *Hs* for MYI | 0.57 | 0.15 |

This result is in direct comparison to the retrieval carried out in the manuscript (Figure 7.c and d of the manuscript). Table 1 (above) shows the direct comparison of the $R^2$ for the least squares fit under the constraint of slope=1. It is shown that with the integration of SMOS TB, the quality of retrieval is improved mainly for *Hs*. For *Hi*, we consider the dominant factor is the value of freeboard, and the error in the retrieved *Hs* in Figure 3 (above) does not have big impact over the fitting for *Hi*.

*b) The retrieval as performed in this study is not very representative for an "actual" retrieval:*

*1. Only 50% of the available SMOS grid cells are used for the analysis, based on the criterion that "the error" (do you mean RMSE here?) between simulated and SMOS observed TB is < 1.5 K (as compared to 3.1 K for all SMOS cells). In a "real" retrieval*

*situation where we do not have the information from OIB (i.e. hi, hs, and surface temperature) to simulate TBs, how can we identify these cases where simulated and SMOS-observed TBs differ more?*

The authors would like to clarify that: (1) the retrieval with half of the points is not an inherent limitation of the proposed method (shown below in details), (2) the RMSE of TB for all the cells is 3.1 K, while the value of 1.41 K is the RMSE of cells with better altimetry (OIB) coverage, (3) the sufficient spatial sampling of altimetry (e.g., OIB) potentially results in better retrieval results, which can be detected through other methods or data (discussed below). Since there is inherent discrepancy between the sea ice cover that generates the SMOS TB observations and that scanned by OIB (or any type of altimetry), we carry out the analysis of the TB error and its relationship to the coverage of OIB. To simplify the analysis, we use the OIB sample count ($M$) as the criterion for the spatial coverage of OIB. OIB campaigns include certain areas with extensive fly-over instead of a single line scan, and the effective sample count $M$ is large for the corresponding cells. Figure 4 (below) shows the decrease of RMSE in TB (modeled v.s. SMOS) with the increase in $M$, analyzed over all available OIB data. For cells with larger $M$ (i.e., over 95-percentile for $M$), the RMSE of TB drops from 3.1 K (for all cells) to 1.41 K, which is also mentioned in the manuscript (pg. 6, l. 5-6). Therefore, for the analysis in Section 4, we carry out retrieval for all cells with modeled TB within 1.5 K of the observed TB.

[Figure]

Figure 4. The relationship of RMSE of TB to OIB sample count.
For each sample count bin, the statistics of RMSE of TB are computed.
Shaded area covers the 5-th and 95-th percentile of the absolute TB error.

We have also carried out retrieval with ALL points (with SMOS TB and global value of $s$). The verification for Hi and Hs are shown in Figure 5 (below). The quality of retrieval witnessed a slight drop when all cells are involved ($R^2$ of 0.89 to 0.86 for *Hi*, and 0.64 to 0.56 for *Hs*). Still, the both parameters can be retrieved.

[Figure]

(a) *Hi*.        (b) *Hs*.

Figure 5. Verification with all points.

As demonstrated, the lack of sufficient spatial sampling of altimetry scanning plays an important role in the forward L-band model as well as the retrieval, and it is also very important to traditional satellite altimetry. With respect to the

referee's question regarding the detection of potential TB errors, the authors consider utilizing methods and data that can support the detection of sub-cell variability of the sea ice cover (e.g., within 40km for SMOS). High-resolution remote sensing such as high-frequency radiometry (89 GHz channel from AMSR-E/2) or visible/infrared (onboard MODIS) serve as candidates. How they can be used for the study of representativeness of altimetry scans and the combined retrieval with L-band data, remains an important direction of future study.

*2. In the retrieval here, TBs are simulated taking the surface temperature from OIB measurements. In a "real" retrieval situation, one would have to use other sources for surface temperature, which have different spatial and temporal resolutions; they could be (and most likely are) measured with many hours time shift! And as surface temperature has been shown to have a huge influence on L-band TB (see e.g. Maaß et al., 2013) and surface temperature can vary on short time scales, the results can be very different when using other temperature information. This is shortly mentioned at the very end of the manuscript, but it is not stressed how much this can influence the retrieval performance.*

As pointed out by the referee, the surface temperature is generally not available for the retrieval. There are several third-party datasets that serve as candidates, including those based on remote sensing (such as MODIS and AMSR-E/2) and reanalysis data (e.g., ERA-Interim). Currently we are considering the retrieval on the daily basis. For example, the MODIS surface temperature field (of about 1 km of spatial resolution) on the same date of the SMOS TB is regridded to the retrieval scale of EASE grid (12.5 km), and further used for the retrieval. In a separate study, the MODIS data is adopted for the retrieval with radar altimetry (from CryoSat-2) and SMOS TB on a daily basis, and the retrieval generates good results (not shown here).

As noted by the referee, rapid change of surface temperature is present for the Arctic. However, we consider the rapid warming of the sea ice covered region is rare during winter (when altimetry is available), and mainly associated with sea ice leads, involving fast sea ice growth. As noted in Zhou et. al. (2017), the effects of open water and (refrozen) leads can be integrated in modeling of TB using third-party data (e.g., sea ice concentration and lead maps), by treating it as a small-scale variability of the sea ice cover.

*3. The retrieval is done for freeboard measurements from the OIB campaign's laser altimeter. It is not discussed how the freeboard data from this altimeter compare with satellite based laser altimeters, which are given as the target for an "actual" retrieval.*

The authors thank the referee for the comment on relationship with actual retrieval with satellite altimetry. The resolution of OIB is 40 m in the direction of the flight track, with the swath (across track) of about 100 m. As a comparison, for ICESat (or the to-be-launched ICESat-2) the resolution for each altimetry scan is about 70 m in diameter, with about 170 m between scans in the direction of flight track. Without considering the availability issues of laser scans (due to penetration problems caused by clouds), there exists resolution discrepancy between OIB and actual satellite measurements by ICESat.

[Figure]

(a) FYI.  (b) MYI.

Figure 6. Scaling of *s* derived from OIB data. PDFs of *s* are shown.

When working with satellite data, the characteristics of the covariability on the spatial scale of satellite altimetry has to be deduced. With respect to the covariability in the retrieval study, the issue is to compute the proper value of $s$. We have carried out analysis by manually coarsening OIB's data to various spatial scales (by computing mean values for $Hs$ and $FB_{snow}$ with adjacent scans), and the results are shown in Figure 6 (above). There is a slight shift of $s$ to smaller values for both FYI and MYI, when coarsening the OIB scans. Since ICESat(-2) features approximately 70 m spatial resolution, we intend to use the interpolated value of $s$ between 40 m and 80 m for the potential retrieval with ICESat-2 and the concurrent satellite campaigns, such as WCOM.

*c) With the current presentation of results I am not convinced that this is more than "playing around" with the relationship between ice thickness and snow depth (as observed during the OIB flights), which is taken advantage of. One would expect a relationship in that older ice is on average thicker and has had more time to accumulate snow on top, which leads to on average thicker snow on thicker ice (considering larger scales). And of course snow depth and freeboard are correlated as snow depth and ice thickness (together with snow and ice densities) determine the freeboard. Thus, I would not consider it as an "result" that you found a correlation between them. One interpretation (assuming that the densities of ice and water are relatively constant) is that the obtained R^2 value shows the fraction of the FB that is actually snow (R^2=0.53 would then mean that about 53% of the FB part consists of snow, while the rest is ice). This is also pointed out in the paper by Kwok et al. (2011), which is cited by the authors.*

The authors would like to provide further proof that it is not statistics between $Hi$ (or $FB_{snow}$) and $Hs$ that plays a central role in the retrieval. First, we ignore the covariability by carrying out retrieval with the mean value of $FB_{snow}$. Specifically, for each previous retrieval problem (which involves multiple altimetry measurements and a single SMOS TB), we compute the mean of the $M$ OIB samples of $FB_{snow}$, denoted $\overline{FB_{snow}}$. Then the retrieval is carried out between $\overline{FB_{snow}}$ and TB, and the retrieved $Hi$ and $Hs$ are compared with the mean sea ice thickness and mean snow depth as measured by OIB (denoted $\overline{Hi}$ and $\overline{Hs}$). The results are shown in Figure 7 (below). This idealized retrieval scenario ignores the resolution difference between laser altimetry and L-band SMOS data. It is worth noting that no covariability is incorporated. But still good retrieval is achieved, with $R^2$ for $Hi$ of 0.78 and that for $Hs$ of 0.50.

[Figure]

(a) $Hi$.                      (b) $Hs$.

Figure 7. Retrieval with TB and mean $FB_{snow}$. Dashed (dotted-dashed) line in each
subfigure is the least-squares line (least-squares line under the slope=1 constraint).

Second, we further compare these results with the reported results in the manuscript, which is realistic retrieval involving multiple $FB_{snow}$ measurements from laser altimetry. With the variability (i.e., $M$ samples), the retrieval of $Hi$ and $Hs$ is improved, with $R^2$ rising to 0.89 and 0.64, respectively (see Figure 7.c and d in the manuscript). This indicates that the altimetry sampling provides necessary information for the retrieval accuracy. It also provides further clarification of the role of the statistics during retrieval: (1) the covariability between $FB_{snow}$ and $Hs$ DOES play an important role for the retrievability during the realistic retrieval, which involves relatively high-resolution altimetry scans, as studied in Section 3.2 of the manuscript, and (2) the statistical information DOES NOT play a central role in the general data fusion methodology of L-band TB and laser altimetry.

*Comments B:*

*- Acknowledgements: OIB and SMOS data are not cited correctly here, see e.g. the NSIDC's and ICDC's (University of Hamburg) conditions/suggestions of how to cite the data usage*

According to referee's suggestions, the acknowledgements is revised as below: "*This work is partially supported by National Key R&D Program of China under the grant number of 2017YFA0603902 and the General Program of National Science Foundation of China under the grant number of 41575076. The authors would like to thank the editors and referees for their invaluable efforts in improving the manuscript. SMOS data is provided from Integrated Climate Data Center (ICDC), icdc.cen.uni-hamburg.de, University of Hamburg, Germany, Digital media.* http://icdc.cen.uni-hamburg.de/1/daten/cryosphere/l3b-smos-tb.html. *[Data Accessed: 2017/10/25]. OIB and SSM/I sea ice concentrations data are provided by NASA National Snow and Ice Data Center Distributed Active Archive Center, Boulder, Colorado USA. doi: http://dx.doi.org/10.5067/7XJ9HRV50O57. [Data Accessed: 2017/10/25]. Besides, the authors are grateful to Willmes, S. and Heinemann, G. for the provision of Arctic sea ice lead map.*".

*- p. 3, L7f: "Several recent studies focus on the retrieval of snow depth over thick sea ice, based on L-band passive microwave remote sensing data from SMOS (Tian-Kunze et al., 2012)." -> I am not aware of "several" studies, they are also not given here, and the given reference (Tian-Kunze et al., 2012) is not about retrieving snow depth but (thin) ice thickness.*

The authors recognize that the reference associated with snow depth retrieval is erroneous, and would like to make revision as below: "*The recent study in Maaß et al., (2013) has carried out the retrieval of snow depth over thick sea ice, by utilizing L-band passive microwave remote sensing data from SMOS*".

*- p. 4, L17: As far as I know, "the OIB Level-4 product IDCSI4" that is claimed to have been used for 2012 to 2015 in this study is only available for the years 2009 to 2013... (While the IDCSI2 Quicklook data is indeed available for 2012 to 2015)*

The authors recognize the reference here is incomprehensive and revised as below: "*Therein, the OIB Level-4 product IDCSI4 is adopted (Kurtz et al., 2013) for 2012-2013 and the remaining OIB data for 2014-2015 is from IDCSI2 Quicklook product, which is also available at NSIDC DAAC. Both of these two datasets are 40 m in resolution along the track's direction*".

*- p. 4, L11-13: "...based on Burke et al. (1979). ... An adapted version of the model was adopted by Tian-Kunze et al. (2014) and ..." -> This is not correct. Tian-Kunze et al. (2014) use another (simpler) approach, which is originally based on a paper by Menashi et al. (1993). (also: "adapted version... was adopted" sounds strange)*

The authors recognize that this is a mistake in the manuscript. The correct reference should only include Maaß et al., (2013a), which adopted the Burke's model. The whole sentence is then corrected as: "*In Maaß et al. (2013b), this model is applied to sea ice and further used for the retrieval of snow depth over thick sea ice*".

*Comments C:*

*- p. 2, L17: "schematic view of remote sensing of sea ice" -> this seems a bit exaggerated to me, the figure mainly shows the definitions of snow and ice thickness and freeboard...*

According to the suggestions of the referee, the sentence at p.2 L17 and the caption of Figure 1 are revised as follows: "*Figure 1 shows the various parameters related to satellite based laser altimetry and L-band passive radiometry for the sea ice cover*", and "*Sea ice parameters in the active and passive remote sensing of the sea ice cover, including sea ice thickness (hi), snow depth (hs) and snow freeboard (FBs)*".

*- p. 2, L28-29: Not everyone knows what "its adapted version for ... FYI" is*

The sentence is revised as follows: "*... climatological snow depth in Warren et al. (1999) for multi-year sea ice (MYI) and halved for first-year sea ice (FYI)*".

*- p. 3, L9: Here, "near realtime" observations are compared with altimetry "which can only achieve basin coverage on the scale of about one month" -> It would be helpful to add that SMOS provides not only "near realtime" data but also "an almost daily coverage of the polar regions".*

The sentence is revised as follows: "*SMOS provides full coverage of polar regions on a near real-time (daily) basis, observations with full coverage of polar regions. It has great advantage over satellite altimetry ...*".

*- p. 3, L13: "Despite the limited coverage..." -> strange wording/argumentation*

The sentence is revised as follows: "*Although airborne remote sensing methods have limited spatial and temporal coverage, campaigns such as ...*".

*- p. 4, L7: "due to the limitation of satellite's orbital parameters, the inherent resolution is about 40 km" -> The resolution is determined by the antenna size, the frequency and the interferometry ("aperture synthesis") principle, not "orbital parameters".*

The authors recognize the correction by the referee, and the sentence is revised as: "*However, due to the limitation of satellite's antenna size, the effective resolution of L-band radiometer onboard SMOS is about 40 km.*"

*- p. 4, L25-26: "we consider OIB measurements in the adjacent 3x3 cells ... of equal contribution" -> Maybe mention at least that this is an approximation (the contributions are actually not equal, see the SMOS "antenna gain function")*

According to the referee's suggestion to clarify the description, the authors have made two modifications. First, a sentence is added in the SMOS data description in Section 2.1: "*The gridded SMOS TB data field is generated from multiple snapshots within a day, with each snapshot involving multiple incident angles and spatially varying gain*". Second, the sentence mentioned by the referee is revised as "*However, due to the inherent resolution of SMOS is about 40 km and the daily gridded field is used in this study, we approximate the correspondence of OIB and SMOS TB by considering OIB measurements in the adjacent 3x3 cells (the red segment in Figure2) of equal contribution to the SMOS TB at the central cell (the one bounded by thick blue lines in Figure 2)*".

*- p. 4, L30-31: "However, for certain segments of the OIB tracks, there exists extensive scanning which corresponds to a much larger value of M." -> I think it would be better and more precise to state the range of encountered M values (e.g. giving minimum, maximum and mean).*

Figure 8 (below) shows the distribution of *M* for all available OIB data. The mean value of *M* is about 700. The referee is also kindly directed to Figure 4 (above) for the relationship between *M* and the RMSE of TB. Revision to the manuscript is also made to reflect the statistics of *M*.

[Figure]

Figure 8. Distribution of OIB sample count ($M$).

*- p. 5, L3-5: "The purpose of these treatments is to rule out the factors that may compromise the quality of the OIB samples and allow focus on the discussion of the retrieval algorithm." -> However, these conditions that were excluded here do not only influence the OIB data quality but will probably also make a potential retrieval with SMOS data more difficult and should be discussed somewhere.*

The authors would like to point out that, indeed open water and (refrozen) leads can have profound effect on the overall TB, indicated by Kaleschke et al. (2010) and Tian-Kunze et al. (2014), and also studied in Zhou et al. (2017). Specifically, in Zhou et al. (2017), their effect is integrated into the radiation model by considering them (i.e., open water, leads) as a sub-scale mixture of the sea ice cover. Besides, a thermodynamic model for the diagnosis of ice thickness within the leads is adopted. For the basin-scale retrieval, these factors definitely should be considered, by using third-party data as indicators. But for the current study, we focus on the retrieval methods, therefore their effects are not considered. Outlook to how their effects should be accounted for is now included in Section 5 in the revised version of the manuscript.

*- p. 5, L16: Reference to used radiation model should be given already here.*
*From the given reference it is not clear how the authors of that paper have "reformulated the model to include multiple layers for sea ice and snow" (Zhou et al, 2017). If several ice layers are used in the model, the higher order reflection terms should be considered. I did not find a statement on this...*

Following the suggestion of the referee, the authors add the reference to Zhou et al. (2017) to this sentence. The work for the study of radiation model is now currently available online (see Zhou et al. (2017)). The non-coherent model from Burke et al. (1979) and its adaptation to sea ice in Maaß et al. (2013a) is the basis for the study in Zhou et al. (2017). Since much higher overestimation of TB was witnessed for MYI, in Zhou et al. (2017) the drainage of salinity in the top part of the MYI is accounted for, by using a multi-layer formulation of the model. We provide a model description document as a supplement to the revised manuscript.

The original model in Burke et al. (1979) ignores the high-order terms for reflection (beyond $2^{nd}$ order). As reported in Maaß et al. (2013b), there exists different behavior of the multi-layer of Burke's model and the multi-layer coherent Ulaby model. However, two aspects that are relevant to the current study: (1) as compared with the drainage of top-layer salinity, the multi-layer formulation induces a smaller change as compared with the single-layer formulation (adopted by Maaß et al. (2013a)), and for MYI, the integration of salinity profile resulted in a much better fit of modeled TB to SMOS observation, as shown in Zhou et al. (2017); and (2) as shown in Figure 3.c of the manuscript, the potential of retrieval lies in that the retrieval is carried out over constant freeboard lines, and even if TB saturates (beyond 3 m for *Hi*) there still exists good sensitivity for retrieval.

*- p. 7, Eq. 3: Maybe better to use "arctan(...)" because tan^-1(...) could also be interpreted as 1/tan(...)*

Corrected according to the referee's suggestion. All other cases including "*tan^-1*" in the manuscript are corrected to "*arctan*" as well, in order to avoid misinterpretation and ensure consistency.

*- p. 8, L5 - p. 9, L19: First you write about "scanning of alpha" without explaining it at all, then you present the results shown in Fig. 5. Then you explain the "scanning of alpha" procedure and finally refer to Tab. 2. This is very confusing. The scanning of alpha should be explained first. It would also be very interesting to see what values alpha takes in this procedure. Are they similar for the individual retrievals? Are they spread over a large range? Do the results shown in Fig. 5 and Tab. 2 belong to the same analysis?*

According to the suggestions of the referee, we have made revisions to the manuscript: (1) first we include the description of *alpha* and the scanning of it in Section 3.2 (on p. 9), (2) in Section 4, the scanning of *alpha* is introduced in the retrieval process in more clear way. Figure 1 and 2 of this response document also provide the statistics of *alpha*. As shown, with OIB data, there exists a large range of possible values for *alpha*, which is also indicated by the retrieval process (which generates the value of *alpha* as well). In Figure 9 (below), for the typical distribution of $FB_{snow}$ for both FYI and MYI (subfigure a), the scanning of *alpha* and the resulting mean *Hs* is shown (subfigure b). Since the value of *s* is smaller than 1, there exists no inundation, which is consistent to the rarity of such events in the Arctic. Besides, indeed, Figure 5 and Table 2 of the manuscript belong to the same analysis.

[Figure]

(a) $FB_{snow}$ distribution.            (b) Mean *Hs*.

Figure 9. Scanning of parameter $\alpha$ and the corresponding mean snow depth (*Hs*).

*- p. 10, L3-6: "There is minor increase in quality (0.91 versus 0.89 and 0.65 versus 0.637) and a relatively large gap to the "ideal" case. This indicates that the uncertainty (or error) in TB and radiation models plays an important role in affecting the quality of the retrieval.*

*The uncertainty of TB may arise from that of the radiation model, as well as the mismatch between the altimetry and passive microwave remote sensing" I think the very similar results for simulated and SMOS TBs cannot lead to these conclusions about the radiation model or the TB measurements! Even if (theoretically) the radiation model gave completely unrealistic TB values: If you do the hs/hi-retrieval by comparing this radiation model's TBs (for different hs,hi values) with this same radiation model's "true" TB (for the "true" hs,hi values), the difference in retrieved and "true" hs,hi will originate from other assumptions used in the retrieval (here: assumption of Eq. 3, choice of s-value, choice of thresholds, ...) or from the ill-posedness of the problem (TB ambiguities in the model, which can also exist in reality) but not the quality of the model to represent the "real world"/SMOS (because you are comparing it with its own output! You are within its "ideal model world"). In contrast, an existing difference between using SMOS and simulated TBs may contain information on the radiation model's performance to simulate SMOS TBs (and also on the effect of the spatial mismatch between altimeter and satellite measurements).*

*As far as I can see, the difference between using global and local s values tells you something about how good the global s value approach is. Here (with the results for simulated and SMOS TBs being very similar), THIS (=using different s values) is where the "relatively large gap to the 'ideal' case" seems to come from!*

After careful re-check of the manuscript and the results, the authors have discovered a mistake in the manuscript, and would like to sincerely apologize for the error and potentially misleading the referee. The retrieval result with $R^2$ of 0.91 for **Hi** and 0.65 for **Hs** is attained with the local value of **s** and SMOS TB, and NOT with the global value of **s** and the modeled TB. In the manuscript (pg. 10, l. 1-3), the correct sentence should be: "*Furthermore, if the retrieval is based on: (1) observed TB from SMOS, and (2) the locally fitted value of s, the $R^2$ values for the fitting are 0.91 and 0.65 for sea ice thickness and snow depth respectively, with virtually no change in the fitting lines (not shown)*". For the sake of completeness, we have also carried out retrieval with the other combination of modeled TB and global value of **s**, and the $R^2$ values are 0.96 and 0.84 for **Hi** and **Hs**, respectively. These values are very close to the "ideal" retrieval case (which involves modeled TB and the local value of **s**). All the results for fitting (for all 4 combinations) are shown in Figure 10 (below). These results indicate that the difference between modeled and observed TB is indeed a major source of the retrieval error, especially for **Hs**, which is a "correctly" stated sentence in the manuscript (i.e., the sentence the referee mentioned).

[Figure]

Figure 10. Comparison of different configurations in retrieval. Each circle represents
the retrieval with the specific TB and the value of **s**. The bottom circle represents
the realistic retrieval scenario, and the top one represents the "ideal" scenario.
The fitting to observations ($R^2$) for both **Hi** and **Hs** are shown accordingly.

The authors would like to emphasize again that the even if with the mean freeboard and TB, both **Hi** and **Hs** can be retrieved (shown in Figure 7 above). However, when the information of altimetry samples is incorporated, better retrieval can be achieved for both **Hi** and **Hs**. For the retrieval which accounts for the resolution difference between L-band TB and laser altimetry, covariability plays an important role in the retrievability.

*- Fig. 5 would benefit from a legend and/or annotations of some of the lines, it is hard to remember from the figure caption what each of the 7 lines represents...*

Legends are added to each of the subfigures of Figure 5 for better readability. The updated figure is available in the revised manuscript. They are also reproduced below in Figure 11.

[Figure]

(a) Scenario I

(b) Scenario II

(c) Scenario III

(d) Scenario IV

(e) Scenario V

Figure 11. Updated Figure 5 of the manuscript.

*Comments D (not a complete list!):*

*- Usage of "etc" is not very precise, for example on p. 2, L5 & p. 2, L20 & p. 4, L11 & p. 5, L34*

The authors have made revisions to the manuscript for corrections and more precise use of "etc", including those pointed out by the referee. Unnecessary uses of "etc" are deleted.

*- p. 2, L5-6: "there is rapid" -> "there has been rapid"*

Corrected.

*- p. 2, L24: "hence limited spatial coverage" -> not a complete sentence*

Corrected as: "*..., so they are limited in terms of spatial coverage*".

*- p. 3, L6: "researches...obtain snow depth" -> strange wording*

This sentence is revised by segmenting into three parts: "*The retrieval of snow depth with passive microwave satellite remote sensing has been carried out in various studies. In Comiso et al., (2003), multi-band data from AMSR-E are utilized, but only for FYI. Maaß et al., (2013b) explored the retrieval of snow depth over thick sea ice with L-band data from SMOS*".

*- p. 3, L10: "requires the prerequisite" -> either "requires" or "prerequisite"*

Corrected as follows: "*However, the sea ice thickness is required for the retrieval*".

*- p. 3, L25: "achieves successfully retrieval" is not a correct expression*

This sentence is revised as follows: "*..., we demonstrate that the proposed algorithm can simultaneously retrieve both sea ice thickness and snow depth, ...*".

*- p. 3, L26: "correspond" -> "corresponds"*

Corrected.

*- p. 4, L20: "Temporally, the date of each OIB campaign is located, and the SMOS TB data from the specific date is attained for the combined retrieval." -> An example for a case where the readability could be improved. This sounds like a complicated way of saying something like: "OIB and SMOS data from the same day are taken."*

Corrected as indicated by the referee as follows: "*OIB and SMOS data from the same day are taken.*"

*- p. 4, L23-24: "due to the inherent resolution of SMOS data is about 40km, therefore..." is not a correct expression*

The sentence is corrected as follows: "*However, since the inherent resolution of SMOS data is about 40 km, even if the SMOS data product is provided on the 12.5 km resolution (small blue cells in Figure 2), we consider OIB measurements in the ...*"

*- p. 4, L26: "the 9 cells covers" -> "the 9 cells cover"*

Corrected.

*- p. 4, L28: "the total area the contributes" -> "the total area that contributes"*

Corrected.

*- p. 6, L28 - p. 7, L7: This part is hard to understand...*

The authors would like to explain that: the purpose of this paragraph is to introduce the protocol in the covariability analysis. For clarity, the whole paragraph is revised as follows:

"*For the covariability between $FB_{snow}$ and $Hs$, we choose the native resolution of the OIB product (40 m) as the spatial scale for analysis. Each TB corresponds to multiple (M) OIB samples, with each sample containing the measurement for both $FB_{snow}$ and $Hs$. We divide these samples into $FB_{snow}$ bins, with each bin covering 5 cm. In total there are 30 bins, covering the range of 0 to 1.5 m. For samples in each bin, we compute the percentiles and the mean value of $Hs$. Figure 4.a shows the mean $Hs$ and the +/-1 standard deviation range and their relationship with $FB_{snow}$, for 4 representative TB points. Furthermore, we carry out least squares fitting (weighted according to sample count in each bin) between mean $Hs$ and $FB_{snow}$. Among all available TB and OIB data, there exist statistically significant positive correlation between $Hs$ and $FB_{snow}$ for over 90% of all points. The values of $R^2$ are in the range of 0.06 and 0.89 (95% percentile), with the mean value of $R^2$ of 0.53. This indicates that there exists consistent covariability between snow depth and snow freeboard across Arctic sea ice cover.*"

- p. 8, L1-2: "For hs > 0, there will be inundation due to: FB< hs." -> Do you mean: "For hs> 0, there will be inundation for values hs>FB."?

The referee's guess is right. For the sake of clarity, this sentence is revised as: "*For any sample of $FB_{snow}$, if the value of freeboard is smaller than the current value of $hs$, in order to avoid inundation, the snow depth for this sample is assumed to be the same as $FB_{snow}$*".

The authors would like to express sincere thanks to the referee for the insightful comments and invaluable suggestions to the manuscript. Revisions to the manuscript are made according to the suggestions. We hope that through the reply and the revised manuscript, the idea and results are now better conveyed to the referee. We would also like to answer any further questions and comments from the referee.

**References:**

[revised manuscript text omitted]
 fittings with observations in Untersteiner (1964) and Yu and Rothrock (1996), the thermal conductivity is defined as:

$$k_{ice} = 2.034\,Wm^{-1}K^{-1} + 0.13\,Wkg^{-1}m^{-2}\frac{S_{ice}}{T_{ice} - 273.15}$$

$$k_{snow} = 0.31Wm^{-1}K^{-1}$$

In this study, we consider the change of $k_{ice}$ within the sea ice of minor effects, and use a bulk value for $k_{ice}$, resulting in a linear temperature profile within the sea ice. This bulk value is determined by the bulk value of $S_{ice}$. The temperature profile is assumed to be continuous through the media interfaces, and ice temperature is assumed to equal the snow temperature at the snow–ice interface. Given $T_{surf}$ which may be derived from other observations (such as MODIS), the bulk ice and snow temperatures $T_{ice}$ and $T_{snow}$ can be written as:

$$T_{ice} = T_{water} + \frac{1}{2}K(T_{surf} - T_{water})k_{snow}h_{ice}$$

$$T_{snow} = \frac{1}{2}(T_{water} + T_{surf} + K(T_{surf} - T_{water})k_{ice}h_{snow})$$

where $h_{ice}$ is the sea ice thickness and $h_{snow}$ is the snow depth and $K = (k_{snow}h_{ice} + k_{ice}h_{snow})^{-1}$. Since a bulk value is adopted for both $k_{ice}$ and $k_{snow}$, given any $T_{surf}$, the temperature profile is linear within the snow cover, as well as the sea ice. Then, the temperature

of each layer of the sea ice cover can be computed .

For the salinity, sea ice type is considered with differentiation between MYI and FYI. For FYI, the salinity is assumed to be homogeneous in the vertical direction, and equal the bulk salinity as prescribed by the sea ice thickness. The bulk salinity for FYI is in turn adapted from the multi-linear structure in Cox and Weeks (1974), and defined as follows (where $S_{ice}$ is in ppt):

$$S_{ice} = 6.08 * e^{(-5.81*h_{ice})} + 7.409 * e^{(-0.5228*h_{ice})} + 1.5$$

With the deepening of the FYI sea ice cover, the bulk salinity decreases, and its minimum value is kept above 1.5 ppt. On the other hand, for MYI, in order to reflect the effect of brine drainage and flushing during the melt season, a vertical salinity profile is adopted following Schwarzacher et al., (1959). For the $k$-th sea ice layer, the mean salinity ($S_{i,k}$) is prescribed as:

$$S_{i,k} = \frac{1}{2} S_{max}[1 - \cos(\pi z^{a/(z+b)})]$$

where $z$ is the normalized vertical coordinate with respect to sea ice thickness (starting from 0 on the ice surface to 1 on the ice bottom) and $z = (k - 1/2)/N$, $N$ is number of ice layers, and $S_{max} = 3.2\ ppt$, $a$=0.407, $b$=0.573 which are the fitted parameters from in-situ MYI salinity observations. Therefore, for MYI, the sea ice salinity ranges from 0 at the top of the surface ($z = 0$) to $S_{max}$ at the bottom ($z = 1$). The sea water salinity is fixed at constant $33\ g/kg$.

**3. Radiative properties**

The radiation model describes the radiation emitted from snow cover, sea ice and sea water, the brightness temperature (TB) at the top of atmospheric ($TB_{TOA}$) can be described as (Maaß et al., 2013b):

$$TB_{TOA} = (1 - c) * (TB_{water} + (1 - e_{water}) * TB_{cosm}) + c * (TB_{ice} + (1 - e_{ice}) * TB_{cosm}) + \Delta TB_{atm}$$

where $c$ is sea ice concentration, $e_{ice}$ and $TB_{ice}$ the emissivity and TB of sea ice, $e_{water}$ and $TB_{water}$ are the emissivity and TB of sea water, $TB_{cosm}$ is cosmic microwave background radiation, which can be considered as uniform and constant (2.7K). $\Delta TB_{atm}$ is TB from atmospheric contribution ranging from -0.36 K and +5.67 K. $e_{water}$ is from the Fresnel equations in different directions of polarization (Ulaby et al., 1981) and $e_{ice}$ is a function of parameters such as polarization, incidence angle, sea ice thickness, temperature, density, salinity, surface roughness, snow depth and temperature, etc.

Based on Maaß et al., (2013a), permittivity of snow ($\epsilon_{snow}$) is determined by a polynomial fit obtained from measurements at microwave frequencies ranging between 840 MHz and 12.6 GHz (Tiuri et al., 1984) as follows:

$$\epsilon_{snow} = (1 + 0.7\rho_{snow} + 0.7\rho_{snow}^2) + i$$
$$* \left(1.59{\times}10^6{\times}(0.52\rho_{snow} + 0.62\rho_{snow}^2){\times}(f^{-1} + 1.23{\times}10^{-14}\sqrt{f})e^{0.036T}\right)$$

where $\rho_{snow}$ is the relative density of snow (compared to water), $T$ the temperature of snow in degrees Celsius and $f$ the microwave frequency. It should be noted that $\epsilon_{snow}$ depend on snow wetness, which is note considered by the model. Permittivity of sea ice ($\epsilon_{ice}$) is confirmed by brine volume fraction ($V_b$) using empirical relationship in Vant et al., (1978):

$$\epsilon_{ice} = a_1 + a_2 V_b + i * (a_3 + a_4 V_b)$$

where $V_b$ is given in ‰, and the values of $a_1$, $a_2$, $a_3$, and $a_4$ following Kaleschke et al., (2010). Similar to Maaß et al. (2013a), for the permittivity of sea water ($\epsilon_{water}$), empirical

relationship from Klein and Swift (1977) is adopted and permittivity of air ($\epsilon_{air}$) is assumed to be 1. The brine volume fraction $V_b$ can be expressed in the following (Cox and Weeks, 1983):

$$V_b = \frac{\rho_{ice} S_{ice}}{\rho_{brine} S_{brine}(1+k)}$$

where $S_{ice}$ is the ice salinity, $\rho_{ice}$ the ice density, $S_{brine}$ the brine salinity and $\rho_{brine}$ the brine density. $\rho_{brine}$ can be fitted with $S_{brine}$ (Cox and Weeks, 1983):

$$\rho_{brine} = 1 + 0.0008 * S_{brine}$$

where $S_{brine}$ is in ‰. Then the following equation is adopted to relate $S_{brine}$ with $T_{ice}$ (Vant et al., 1978):

$$S_{brine} = a + b * T_{ice} + c * T_{ice}^2 + d * T_{ice}^3$$

where $T_{ice}$ is in °C and $a, b, c, d$ are fitted parameters in Vant et al., (1978). These polynomial approximations agreed well with the experimental data of Zubov and Nikolaĭ, (1963).

Also, $\rho_{ice}$ can be expressed by ice temperature ($T_{ice}$: °C) in Pounder (1965):

$$\rho_{ice} = 0.917 - 1.403 * 10^{-4}\, T_{ice}$$

Therefore, $V_b$ can be expressed as a function of $\rho_{ice}$, $S_{ice}$ and $T_{ice}$.

As derived model from Burke et al. (1979), the radiation model is a non-coherent model. However, the effect of non-coherency is considered to be mitigated by several factors. First, since with the SMOS observations, there exists large variability of both sea ice thickness and snow depth within the typical resolution of 40 km. Second, multi-angle mean of SMOS TB further introduces a range of integration path of radiations. These factors would enable the use of non-coherent model in this study (RMS of **Hi** variation larger than 1/4 of L-band wavelength, as indicated in Kaleschke et al., (2010)). The multi-layer treatment of the sea ice also explored in Maaß et al. (2013b). With treatment of the salinity profile in MYI (i.e., salinity drainage in the top layers), the modeled TB is more consistent with the SMOS TB, as studied in Zhou et al. (2017).

**4. Modeled TB under typical sea ice parameters**

Under typical winter Arctic conditions (surface temperature is $-30°C$), simulated brightness temperature (TB) over different sea ice type from reformulated radiation model shows in Figure S1, along with snow freeboard contour.

[Figure]

Figure S1. Simulated sea ice surface TB based on reformulated radiation model over FYI (a) and MYI (b). Colored lines are snow freeboard isolines.

**5. Supportive information for the verification with OIB and SMOS data**

Verification is carried out between OIB data and SMOS, with specific attention to the effect of better OIB sampling. Due to the resolution difference between SMOS and OIB (or any type of satellite altimetry), we consider multiple ($M$) OIB samples correspond to a single SMOS TB. Figure 2 of the manuscript provides a corresponding relationship between the two. Figure S2 shows the distribution of $M$, using all OIB data. The mean value of $M$ is about 700. The RMSE of TB (modeled v.s. observed) is 3.1 K for all available OIB data (see also Figure 3.b of the manuscript). If we further limit the computation of RMSE to the points with large values of $M$ (95-percentile for $M$, corresponding to areas with good OIB coverage), the RMSE drops to 1.41 K. Figure S3 shows the relationship of RMSE of TB and $M$. As shown, there exists drop in both RMSE and the maximum error of TB with better spatial coverage of OIB.

The integration of lead information is explored in Zhou et al. (2017), which effectively reduces the overestimation of TB as caused by refrozen leads or open water.

[Figure]

Figure S2. Distribution of OIB sample count ($M$).

[Figure]

Figure S3. The relationship of RMSE of TB to OIB sample count ($M$).
The statistics of RMSE of TB are computed for each sample count bin (each of 100).
Shaded area covers the 5-th and 95-th percentile of the absolute TB error.

**Part 2: Statistics for the covariability between *FB$_s$* and *hs**

[Figure]

(a) ***alpha***.        (b) ***beta***.        (c) ***s***.

Figure S4. Distribution of alpha, beta and s for FYI. 40-meter resolution (OIB) data
are used for computing the value of each parameter on the scale of 37.5 km
(i.e., approximately native resolution of SMOS TB)

[Figure]

(a) ***alpha***.        (b) ***beta***.        (c) ***s***.

Figure S5. Distribution of alpha, beta and s for MYI. Specifications are the same as Figure S4.

[Figure]

(a) FYI.             (b) MYI.

Figure S6. Scaling of *s* derived from OIB data. PDFs of *s* are shown.

---

## Author Response (AR2)

**Outline of the document:**

The authors would like to thank the two anonymous referees for the invaluable comments to the revised version of the manuscript, and the editors for the efforts in handling the manuscript. This document contains replies to the comments from the referees (page II to V), as well as the version of the manuscript that contains highlighting to the revisions (page 1 to 32).

**Content:**

**Reply to comments from referee #1:**

The authors thank referee #1 for the constructive and invaluable suggestions for the revision of the manuscript. The reply and corresponding revisions are listed below for each comment. The referee's comment below is in italic. The revisions are marked out in the highlighted version of the manuscript by cyan highlighting.

*Discussion on error assessment and better explanations of the methodology is required before publishing the paper.*

**Author's response**: according to the two suggestions from the referee, we have made the following major revision to the manuscript.

First, we have added a dedicated subsection (Section 4.2, title highlighted in cyan) in Section 4 that covers the uncertainty estimation for the retrieval. The effect of input parameters (SMOS *TB* and snow freeboard, i.e., *FBs*) is evaluated, as well that of the parameter *s* that characterizes the covariability. Beta Distribution is used to characterize *s*, and the perturbations to it. Through Monte-Carlo simulations that contain random perturbations to these input parameters, it is shown that both *TB* and *FBs* play an important role in the uncertainty of retrieved sea ice thickness (*hi*) and snow depth (*hs*). The effect of *s* is comparatively less important to that of *TB* or *FBs*. Overall, the relative uncertainty of *hi* (or *hs*) is always lower than 25%. Monte-Carlo simulation can also be adopted for future works with satellite data for the uncertainty estimation, since it is not directly computable due to the nonlinearity in the radiation model.

Second, with respect to the suggestion to improve the explanation of methodology, we made further clarification of the retrieval algorithm and its relationship with the theoretical retrieval study in Xu et al. (2017). Specifically, in Section 3, we add an extra paragraph (page 7, line 2-7, marked out by cyan) that: (1) better explains the retrievability from the theoretical study (i.e., no resolution difference or covariability), and (2) provides the proper background for the incorporation of covariability. Other pieces of supportive information in other part of the manuscript are also added (such as Section 4, page9, line 30) to clarify the relationship of the work with Xu et al. (2017).

Based on the suggestions of the referee, we sincerely hope that with these revisions, the completeness and the clarity of the manuscript can be improved.

**Reference:**

Xu, S., Zhou, L., Liu, J., Lu, H., and Wang, B.: Data Synergy between Altimetry and L-Band Passive Microwave Remote Sensing for the Retrieval of Sea Ice Parameters - A Theoretical Study of Methodology, Remote Sensing, 9, doi:10.3390/rs9101079, 2017.

**Reply to comments from referee #2:**

The authors thank referee #2 for the invaluable suggestions for the revision of the manuscript. The reply and corresponding revisions are listed below for each comment. The referee's comments are in italic. Corresponding changes in the highlighted version of the manuscript are marked out by yellow or green highlighting (yellow for minor corrections, while green for relatively more significant revisions).

*Referee #2:*
*Excellent research with new method for fusing data from several sources.*

*A few minor improvements could be helpful:*
*1) A paragraph discussing the potential impact of the fixed snow and ice densities and other assumptions.*

**Author's response**: based on the suggestion of the referee, a paragraph addressing the effect of densities is added in Section 5 (line 16 to 21 on page 13, marked by green highlighting). We note that snow/ice density and other "model parameters" should be considered in terms of their effect on the uncertainty of the retrieved parameters. This can be attained through a Monte-Carlo simulation approach, as in Section 4.2 (which is added according to referee #1's suggestion).

*2) The supplement should be incorporated in the paper as an appendix.*

**Author's response**: according to the suggestion of the referee, the two parts of the supplementary material are now formulated as appendices (A and B) in the manuscript (with the title of each appendix marked by green highlighting). Related modifications are made, including the merging of Figure S1 (in the supplementary) into Figure 3 (of the manuscript), and the addition of explanatory text in Appendix B.

*Some minor editorial points:*

*P1L19: as well the -> as well as the*
**Author's response**: corrected.

*P2L15: The basin-scale observation -> Basin-scale observations*
**Author's response**: corrected.

*P2L16: based on satellite retrieval -> retrieved from satellite data*
**Author's response**: corrected.

*P2L24: hence limited spatial -> hence provide only limited*
**Author's response**: corrected to: "… so they are limited in terms of spatial coverage".

*P2L27: certain estimation -> certain assumptions*
**Author's response**: corrected.

*P2L29: Besides, for -> For*
**Author's response**: corrected.

*P3L1: reason of -> reason for*
**Author's response**: corrected.

*P3L6: and many researches can only obtain -> and existing methods only provide*
**Author's response**: the whole sentence is revised as follows for clarity: "The retrieval of snow depth with passive microwave satellite remote sensing has been carried out in various studies".

*P3L21: decided by -> determined by*
**Author's response**: corrected.

*P3L25: successfully -> successful*
**Author's response**: the part of the sentence is revised as follows: "…, we demonstrate that the proposed algorithm can simultaneously retrieve both sea ice thickness and snow depth, …".

*P4L7: satellite's orbital -> satellite's imaging and orbital*
**Author's response**: the part of the sentence is revised as follows: "…, due to the limitation of the satellite's antenna size, …".

*P4L16: Norwegian Meteorological Service OSI SAF-> the EUMETSAT OSI SAF*
**Author's response**: corrected.

*p4L17: track's direction -> along track direction*
**Author's response**: corrected.

*P4L30-31: there exists extensive scanning -> more extensive coverage was acquired*
**Author's response**: the part of the sentence is revised as: "… there exist certain areas that are scanned more extensively, …".

*P5L11: G H z -> GHz*
**Author's response**: corrected.

*P5L27: FBs should be explained. It was not introduced. Could be done in P5L6.*
**Author's response**: formal introduction of *FBs* is added in Section 2.1, as indicated by the referee.

*P5L27: T B -> TB*
**Author's response**: corrected.

*P5++: F B -> FB*
**Author's response**: corrected.

*P8L21: scanning -> scannings*
**Author's response**: corrected.

*P9L6: the different side -> different sides*
**Author's response**: corrected.

*P9L13++: Why use capital H_bar. h_bar should be OK?*
**Author's response**: corrected for both Hi_bar and Hs_bar into non-capitalized versions.

*P11L5: aon -> on*
**Author's response**: corrected.

*P11L13: AMSR2 -> AMSR-E*
**Author's response**: corrected.

*P11L13: Discuss advantages and disadvantages of using C-band instead of L-band, or using both C-band and L-band with Cryosat RA.*
**Author's response**: according to the suggestion, the authors have added a paragraph in Section 5 (page 14, line 13-20, marked in green). This added content extends the overall discussion in Section 5, and contain the specific considerations involving using C-band instead of L-band with AMSR-E, as indicated. The other aspect involving Cryosat RA is further discussed in the following up paragraph (also added content) in Section 5.

*P11: Please also introduce the possibility of using the combination of Cryosat and SMOS (which are available from roughly the same time period)*
**Author's response**: according to the suggestions of the referee concerning Cryosat RA (this comments and the previous comment), the authors have summarized a new paragraph in Section 5 (page 14, line 21-29, marked in green) for the discussion concerning RA. It is noted that RA and L-band passive data can be combined for retrieval, and due to (similar) resolution differences, the effect of covariability should also be considered for the retrieval.

[revised manuscript text omitted]